# 🌮 TACO: Training-free Sound Prompted Segmentation via Semantically Constrained Audio-visual CO-factorization

**Hugo Malard**
*LTCI, Télécom Paris, Institut Polytechnique de Paris, France*

**Michel Olvera**
*LTCI, Télécom Paris, Institut Polytechnique de Paris, France*

**Stéphane Lathuilière**
*Inria at Université Grenoble Alpes, CNRS, LJK, France*

**Slim Essid**
*NVIDIA, France* [†]

Reviewed on OpenReview: *https://openreview.net/forum?id=Xt9sdzQQlJ*

## Abstract

Large-scale pre-trained audio and image models demonstrate an unprecedented degree of generalization, making them suitable for a wide range of applications. Here, we tackle the specific task of sound-prompted segmentation, aiming to segment image regions corresponding to objects heard in an audio signal. Most existing approaches tackle this problem by fine-tuning pre-trained models or by training additional modules specifically for the task. We adopt a different strategy: we introduce a training-free approach that leverages Non-negative Matrix Factorization (NMF) to co-factorize audio and visual features from pre-trained models so as to reveal shared interpretable concepts. These concepts are passed on to an open-vocabulary segmentation model for precise segmentation maps. By using frozen pre-trained models, our method achieves high generalization and establishes state-of-the-art performance in unsupervised sound-prompted segmentation, significantly surpassing previous unsupervised methods. The code of our experiments is open source[*].

## 1 Introduction

Audio-visual perception has attracted considerable interest due to its practical applications in various fields, especially robotics, video understanding (Chen et al., 2023; Shahabaz & Sarkar, 2024), and recently, audio-visual language modeling (Malard et al., 2024; Chowdhury et al., 2024). A particularly compelling task in this area is sound-prompted segmentation (Hamilton et al., 2024)—also known as audio-visual segmentation (Zhou et al., 2022) —which involves identifying image regions associated with sounds in the accompanying audio. Whereas several prior approaches frame this task as a supervised segmentation problem (Liu et al., 2023b; Gao et al., 2024; Liu et al., 2023a), some explore unsupervised learning, relying solely on inherent cross-modal features rather than ground-truth audiovisual masks. Most of them proceed to aligning audio and visual representations through contrastive learning techniques (Sun et al., 2023; Senocak et al., 2023; Park et al., 2023). Such techniques exploit paired audio-visual data to learn a shared embedding space,

---

[†]Work conducted while at Telecom Paris.
[*]Code available at:
https://github.com/hugomalard/TACO-Training-free-Sound-Prompted-Segmentation-via-Audio-visual-CO-factorization.git

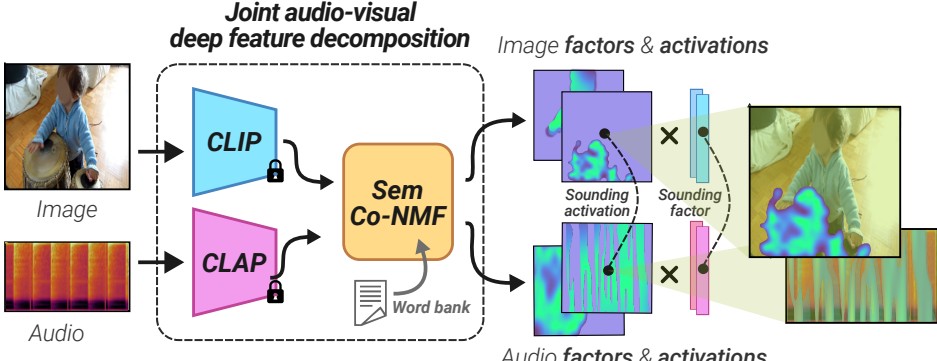

Figure 1: Our method takes a representation of an image and its associated audio as input, decomposing them into a product of 'semantic' factors and (spatial or temporal) activations. This decomposition enables locating parts of the original input corresponding to the concept present in both the image and the audio.

where corresponding audio and visual features are brought closer together, while non-corresponding pairs are pushed apart. The main challenge lies in learning local features that can be used for segmenting audio-related concepts, as videos typically provide global-level alignment (as opposed to local alignment, where each part of the image has a match with a defined part of the audio). Recently, a few works have made significant progress in this direction through the use of pre-trained models. Park et al. (2024) use a pre-trained CLIP encoder (Radford et al., 2021) and audio tokenizer within a relatively complex framework that translates audio signals into tokens compatible with CLIP's text encoder. This is combined with a grounding mechanism, enabling audio-driven segmentation of the image. Similarly, DenseAV (Hamilton et al., 2024) identifies image regions linked to sounds without supervision. Using a DINO image backbone (Caron et al., 2021) and the HUBERT audio transformer (Hsu et al., 2021), it employs multi-head feature aggregation for contrastive learning on video-audio pairs.

These approaches tackle sound-prompted segmentation through carefully designed architectures and dedicated training procedures, which often compromise the model's generalization capabilities and abilities in other tasks. Here, we propose a radically different strategy. Specifically, we ask: *Can sound-prompted visual segmentation be achieved using only pre-trained audio and image models, in an unsupervised manner, by analyzing their frozen feature representations?*

To answer this question, we introduce **TACO**: a *Training-free Audio-visual CO-factorization* approach, which identifies audio and visual tokens that are *co-activated* in the input signal. Leveraging pre-trained CLIP and CLAP (Elizalde et al., 2023) backbones for image and audio, respectively, we introduce a semantically constrained Non-negative Matrix co-Factorization (Sem co-NMF) framework. This co-NMF formulation enables audio-visual correspondence analysis without the need for training. Specifically, using a set of semantic *anchor words* projected into the CLIP and CLAP spaces, we are able to match audio features with visual ones in an interpretable fashion. Through simple inference-time optimization, our framework establishes local correspondence between visual and audio signals, and further performs sound-prompted segmentation in a training-free manner, as illustrated in Figure 1. Furthermore, the interpretable factors identified by our decomposition framework allow us to directly prompt a pre-trained open-vocabulary segmenter, such as FC-CLIP (Yu et al., 2023b), enhancing segmentation quality while remaining zero-shot.

Our unsupervised approach stands out for its training-free paradigm which preserves the generalization capabilities of the pretrained models, as well as the interpretability offered by the NMF framework. Here, *training-free* means that the method does not rely on any dataset, opting instead for sample-specific optimization. This training-free setup holds great potential for multi-task scenarios, enabling a single model to support multiple downstream applications without additional fine-tuning. Additionally, leveraging the robustness of the frozen CLIP and CLAP models enables us to achieve significantly superior performance compared to existing unsupervised training-based methods on established benchmarks. To summarize, our contributions are:

- We introduce **TACO**, a training-free approach for sound-prompted segmentation based on pre-trained deep representations. Our framework leverages the inherent interpretability of matrix factorization, enabling clear, semantically grounded visualization of interactions between segmentation outputs and the concepts identified within the signal.
- We revisit soft co-NMF for audiovisual sound source localization by constraining the semantics of the decomposition and incorporating a novel penalty function. This new decomposition (called Sem co-NMF) enables finding audio-visual correspondences across unaligned sound and image representation backbones.
- Our Sem co-NMF framework is meant to extract both an audiovisual segmentation and encode audiovisual concepts that can be incorporated into a pre-trained open-vocabulary model (namely, FC-CLIP (Yu et al., 2023b)), prompting its decoder with the most activated factors identified by our decomposition system, allowing significant leap in accuracy.
- We rigorously validate our approach through extensive experiments, providing both quantitative and qualitative analyses. We demonstrate the superior performance of our unsupervised (training-free) approach across four datasets and multiple variants of the sound-prompted segmentation task. The code will be released as open-source upon acceptance.

## 2 Related Work

**Audio visual learning**  Audio-visual learning has been widely explored to model correspondences between visual content and temporally evolving audio signals. Early works on audio-visual event localization explicitly study temporal alignment between sound events and visual dynamics in unconstrained videos (Tian et al., 2018a), while subsequent approaches learn shared audio-visual representations through weak or self-supervision for cross-modal grounding and retrieval (Arandjelovic & Zisserman, 2017; Mo & Morgado, 2022a). More recent methods leverage large-scale pre-training to learn unified spatio-temporal multimodal representations, enabling flexible transfer across audio-visual tasks (Gong et al., 2023; Girdhar et al., 2023). Rather than learning audio-visual alignment through training, we leverage frozen pre-trained representations and training-free co-factorization to directly extract local audio-visual correspondences.

**Unsupervised Sound-Prompted Segmentation**  The sound-prompted segmentation task aims to localize the object in an image that emits the sound present in the associated audio. While often referred to as audio-visual segmentation (Zhou et al., 2022) , we adopt the term used by Hamilton et al. (2024) throughout this paper, as it more clearly indicates that only the image is segmented, not the audio. Recent methods primarily utilize cross-modal attention for sound-prompted segmentation (Tian et al., 2018b; Senocak et al., 2019; 2018), often combined with a contrastive loss that aligns audio and image global representations, hypothesizing that this process would implicitly foster the emergence of local alignment. Extending this contrastive learning framework, numerous enhancements have been introduced. These include integrating challenging negatives from background areas (Chen et al., 2021), adopting iterative contrastive learning with pseudo-labels from earlier model epochs (Lin et al., 2023), maintaining transformation invariance and equivariance through geometric consistency (Liu et al., 2022), leveraging semantically similar hard positives (Senocak et al., 2022), and implementing false negative-aware contrastive learning via intra-modal similarities (Sun et al., 2023).
More recently, large pre-trained models have started to be used in order to benefit from their rich data representation. ACL-SSL (Park et al., 2024) relies on the CLIP image encoder as well as a segmentation model CLIP-Seg (Lüddecke & Ecker, 2022), and learns an additional module that projects the audio in the CLIP space. Relying on DINO and HUBERT (further fine-tuned),  Hamilton et al. (2024) trained a multi-head module that identifies the localization of the sound in the image features. Unlike these methods, our framework uses a training-free setup with pre-trained models and matrix decomposition, requiring no training data and preserving the model's generalization abilities. Finally, a few recent methods attempt to perform sound-prompted segmentation in a training-free manner by chain-prompting multiple single-modality models. In particular, Box-Prompt (Yu et al., 2023a) combines CLAP (Elizalde et al., 2023), GroundingDINO (Liu et al., 2024), and SAM (Kirillov et al., 2023) to perform the task. Extending this paradigm to very large models, OpenAVS (Chen et al., 2025) integrates the reasoning capabilities of Large Language Models, such as GPT-4o-mini, with specialized frameworks including Pengi (Deshmukh et al.,

2023), GroundingDINO (Liu et al., 2024), and SAM2 (Ravi et al., 2024). While these approaches achieve good segmentation performance, they suffer from prohibitive computational cost. In contrast, our method focuses on directly decomposing features from the audio and image encoders to discover *audiovisual correspondences*, limiting the computational overhead.

**NMF and Deep Feature Factorization**   Non-negative matrix factorization (NMF) has been applied in diverse fields, including audio source separation (Grais & Erdogan, 2011), document clustering (Xu et al., 2003), and face recognition (Guillamet & Vitria, 2002). Previous research has expanded NMF to multiple layers (Cichocki et al., 2006), implemented NMF using neural networks (Dziugaite & Roy, 2015), and utilized NMF approximations as inputs to neural networks (Vu et al., 2016). More recently, it has been applied to decompose deep neural features, for concept discovery (Collins et al., 2018), interpretability (Parekh et al., 2022; 2024) or audio-visual source separation (Parekh et al., 2017). While it has already been applied to co-factorize audio and visual (handcrafted) features (Seichepine et al., 2014), to the best of our knowledge, this is the first work applying it to deep features for sound-prompted segmentation, and the first to co-factorize representations from different spaces while enforcing semantic matching.

## 3   🌮 TACO: Training-free semantically constrained Audio-visual CO-factorization

Our methodology leverages the capacity of recent large pre-trained models to effectively *localize* patterns within their input signals (Yu et al., 2023b). *Localization* here refers to identifying spatial regions in images and spectro-temporal segments in audio. From this observation, we hypothesize that separately trained audio and visual encoders contain all the necessary information for sound-prompted segmentation without requiring additional training. To achieve this, we propose a method, **TACO**, which identifies concepts that co-activate across audio and image representations exploiting the Non-negative Matrix Factorization (NMF) framework (Lee & Seung, 2000). NMF provides a training-free approach that jointly decomposes each signal into an activation matrix and a concept matrix. The activation matrix, in particular, can be interpreted as a segmentation matrix, highlighting regions of concept alignment.

Before detailing our approach, it is important to note that NMF is a tool that is meant to interpret input matrices composed of only non-negative values. The non-negativity constraints are key to the interpretability offered by the NMF framework. However, deep representations often include negative coefficients. A simple solution is to clip these negative values to zero, ensuring matrix non-negativity. While this may lead to some information loss, our preliminary experiments show that with backbones like CLIP (Radford et al., 2021) and CLAP (Elizalde et al., 2023), representation power remains unaffected for our tasks (further details are in Sec. 4). Section 3.1 provides the necessary background on NMF, and section 3.2 introduces our framework for extracting audio-visual concepts using NMF, while section 3.3 describes our complete solution for audio-source segmentation, which integrates our factorization approach into a pre-trained open-vocabulary image segmenter, enabling fine segmentation in a training-free manner.

### 3.1   Background

**NMF**   Non-negative Matrix Factorization (NMF) (Lee & Seung, 2000) involves decomposing a non-negative matrix $X \in \mathbb{R}_+^{N \times C}$ into two non-negative matrices $U \in \mathbb{R}_+^{N \times K}$ and $V \in \mathbb{R}_+^{K \times C}$ such that $X \approx UV$. In typical machine learning applications of NMF (Gillis, 2014), $N$ represents the number of observations and $C$ denotes the number of (non-negative) features per observation, while $K$ is the rank of the factorization, controlling the number of *factors* or *components* into which the original matrix is decomposed. Given a distance $D$ in the matrix space, NMF can be formulated as the following constrained optimization problem:

$$\min_{U,V \geq 0} D(X|UV). \tag{1}$$

The matrix $V$ is called *factor* matrix, and the $U$ matrix corresponds to the *activations* of those factors.

**Co-NMF**   extends the NMF framework to multiple data views analyzed in parallel. Assuming two data matrices $X_1 \in \mathbb{R}_+^{N \times C}$ and $X_2 \in \mathbb{R}_+^{N \times C}$ containing different modalities (such as audio and visual streams of a video), co-NMF exploits the mutual information shared between these modalities. It assumes that

the different modalities share a common activation matrix $U$. In particular, Yokoya et al. (2011); Yoo & Choi (2011) decompose these matrices into a shared activation matrix $U \in \mathbb{R}_+^{N \times K}$ with view-specific factors $V_1 \in \mathbb{R}_+^{K \times C}$ and $V_2 \in \mathbb{R}_+^{K \times C}$. This formulation forces the activation matrix $U$ to be consistent across both modalities, reflecting the intuition that events occurring in the audio also happen at the same time in the image stream. Although the factors are encoded in different spaces, they should remain the same. Formally, given distances $D_1$ and $D_2$, co-NMF solves the following problem:

$$\min_{U, V_1, V_2} D_1(X_1|UV_1) + D_2(X_2|UV_2), \tag{2}$$

subject to non-negativity constraints on $U$, $V_1$, and $V_2$.

Note that, as there might be some small temporal shift between the audio and image events, enforcing the same $U$ matrix for both decompositions can be overly restrictive. To mitigate this issue, soft non-negative Matrix Co-Factorization (**soft co-NMF**) (Seichepine et al., 2014) introduces two separate activation matrices and includes a penalty term to minimize their dissimilarity. They are obtained by solving the following optimization problem, with a penalty function $P$, and hyperparameter $\beta_p > 0$

$$\min_{U_1, U_1, V_2, V_2} D_1(X_1|U_1V_1) + D_2(X_2|U_2V_2) + \beta_p P(U_1, V_1, U_2, V_2), \tag{3}$$

subject to non-negativity constraints on $U_1$, $U_2$, $V_1$ and $V_2$.

### 3.2 Sem Co-NMF: Revisiting soft co-NMF

This section details our redefinition of soft co-NMF to semantically constrained co-NMF (Sem Co-NMF) and its application in **TACO**.

**Soft co-NMF for token decomposition** We assume access to an input video with a corresponding audio signal. For simplicity, let us initially consider the central frame of the video; we will later extend our approach to handle the entire video, incorporating the temporal dimension. Additionally, we assume the availability of trained audio and image backbones. Feeding the audio signal into the audio backbone produces a matrix $X_A \in \mathbb{R}_+^{N_T \times C_A}$, where $N_T$ is the number of tokens, and $C_A$ is the token dimension. Similarly, the image encoder maps an image to a feature representation $X_I \in \mathbb{R}_+^{HW \times C_I}$, where $H$ and $W$ represent the spatial dimensions, and $C_I$ is the number of channels. Note that the spatial dimensions are flattened, resulting in a 2D matrix compatible with NMF inputs.

In the original formulation, soft co-NMF factorizes visual and audio matrices containing temporal features (i.e. the first dimension corresponds to time). The penalty term is applied to the activations to ensure that concepts align temporally in both audio and image. However, in our case, the first dimension of the image feature corresponds to space (as we aim for spatial decomposition for segmentation), hence the revision of the factorization model.

In our Sem co-NMF scheme, we aim at factorizing $X_A$ into matrices $U_A \in \mathbb{R}_+^{N_T \times K}$ and $V_A \in \mathbb{R}_+^{K \times C_A}$, along with $X_I$ into $U_I \in \mathbb{R}_+^{HW \times K}$ and $V_I \in \mathbb{R}_+^{K \times C_I}$, where $K$ denotes the predefined number of factors. Restricting the decomposition of the $U_I$ and $U_A$ matrices to the interval $[0, 1]$ instead of $\mathbb{R}_+$ provides an interpretable, well-calibrated scale, where each concept is either activated (near one) or inactive (near zero) in each token. Thus, we re-parameterize the problem using the sigmoid function $\sigma$ and optimize matrices $\tilde{U}_A \in \mathbb{R}^{N_T \times K}$ and $\tilde{U}_I \in \mathbb{R}^{HW \times K}$, such that $U_A = \sigma(\tilde{U}_A)$ and $U_I = \sigma(\tilde{U}_I)$ with $U_I \in [0, 1]^{HW \times K}$ and $U_A \in [0, 1]^{N_T \times K}$. Specifically, the $k$-th column of $U_I$ and $U_A$ can be interpreted as the activation of the $k^{th}$ factor for each token. In this way, the Hadamard product between the original tokens and a column of $U_I$ or $U_A$ can be viewed as soft masking, highlighting the portion of the original input that activates the $k^{th}$ factor. To *jointly* decompose the image and audio features, we cannot constrain the activations to be identical, as images encode spatial information and audio captures spectro-temporal patterns. Additionally, constraining the factors $V_A$ and $V_I$ is challenging, as they reside in different spaces. To address this, we introduce a penalty function that ensures semantic proximity between the decompositions.

**Penalty function** Applying soft co-NMF to a specific problem requires careful definition of the penalty term $P$ in Eq. equation 3, as it impacts both the quality of the factorization and its interpretability. To

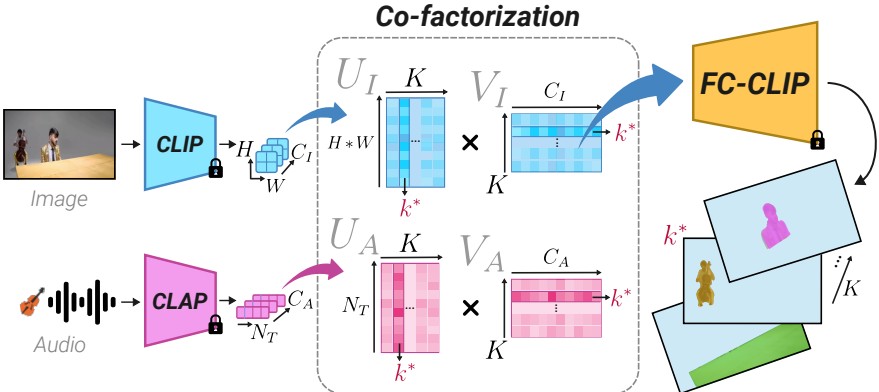

Figure 2: Complete pipeline: both the audio and the image are encoded using their respective encoder and their representations are used to perform the co-NMF. FC-CLIP is prompted using the image factors $(V_I)$ and the segmentation corresponding to the sounding image factor $(V_I^{k\star})$ is kept as the final segmentation.

achieve a semantically consistent factorization of audio-visual observations, $P$ must effectively capture the semantic similarity between the image and audio features. However, since the backbones (e.g., CLIP and CLAP) are not aligned, audio and image factors are encoded in two different subspaces, making $L_1$ or $L_2$ norms unsuitable as penalty functions. Moreover, audio and image representations are not necessarily commensurable: they often differ in dimensionality, precluding direct use of standard distances. To address this, we propose projecting the audio and visual feature representations into a unified feature space to enable direct comparison. Specifically, we estimate aligned semantic descriptors $\mathcal{D}_{I^k}$ and $\mathcal{D}_{A^k}$ for the $k^{th}$ column of $U_I$ and $U_A$, respectively. These descriptors are computed using $J$ *semantic anchors*, $\{b_I^j\}_{j=1}^J$ and $\{b_A^j\}_{j=1}^J$, which lie in the same spaces as $X_I$ and $X_A$. Each anchor pair $(b_I^j, b_A^j)$ represents a shared semantic concept, enabling the estimation of aligned semantic descriptors. Specifically, to estimate aligned semantic anchors, we utilize CLIP (Radford et al., 2021) and CLAP (Elizalde et al., 2023) as image and audio backbones. Since these encoders are aligned with text encoders, we can exploit their respective text embeddings to align identical words across audio and image spaces. Thus, we define $b_I^j = \mathcal{E}_I(t_j)$ and $b_A^j = \mathcal{E}_A(t_j)$, where $\mathcal{E}_I$ and $\mathcal{E}_A$ are the text encoders of CLIP and CLAP, respectively, and $\{t_j\}_{j=1}^J$ represents a predefined word bank.

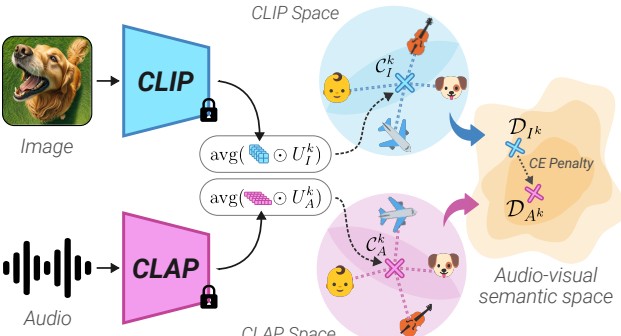

Figure 3: As CLIP and CLAP encode audio and images in different spaces, our method employs semantic anchors to project semantic components (representations soft-masked by $U_I^k$ and $U_A^k$) in an audio-visual semantic space where standard distances can be considered to compute the penalty function.

We now describe the process of estimating the semantic descriptors from the matrices $U_A$ and $U_I$ using the aligned semantic anchors. For simplicity, we detail only the estimation of semantic image descriptors; analogous steps apply to the audio descriptors. First, we introduce *semantic components* $\mathcal{C}_I^k$ associated to each column $k$ of $U_I$. Since $U_I^k$ represents the spatial activation of the $k^{th}$ factor (corresponding to $V_I^k$),

it can be interpreted as a soft segmentation mask for the input $X_I$. Applying these soft-masks to $X_I$ and averaging the resulting representations extracts the concept associated with the $k^{th}$ factor from the image features. The $k^{th}$ image semantic component is defined as:

$$\mathcal{C}_I^k = \text{avg}(X_I \odot U_I^k), \tag{4}$$

where $\text{avg}(.)$ denotes spatial average pooling (i.e. $f : \mathbb{R}^{HW \times C_I} \to \mathbb{R}^{C_I}$) and $\odot$ is the Hadamard product. We define the semantic image descriptor $\mathcal{D}_{I^k}$ by computing the cosine similarity between $\mathcal{C}_I^k$ and each semantic anchor:

$$\mathcal{D}_{I^k} = \left[ \cos(\mathcal{C}_I^k, b_I^1), \ \cdots, \ \cos(\mathcal{C}_I^k, b_I^J) \right]^T \tag{5}$$

where $\cos(.)$ denotes cosine similarity. Intuitively, a descriptor quantifies the similarity of the semantic component with each semantic anchor (illustrated in Figure 3). Note that $V_I^k$ could have been used to play the role of semantic components instead of $\mathcal{C}_I^k$. However, $V_I^k$ is less resilient to variations of the semantic anchors. Indeed, during the NMF optimization, $V_I^k$ can take any range of values, potentially degenerating to values very similar to those of $b_I$, which would be equivalent to performing a simple classification. In contrast, each $\mathcal{C}_I^k$ is inherently constrained to be a weighted average of elements from the original representations, thus discouraging it from merely replicating values from $b_I$. We ablate this choice in Section 4. Audio semantic descriptors $\mathcal{D}_{A^k}$ are computed similarly to their image counterparts. The penalty function is then defined as the cross-entropy between the image and audio semantic descriptors (alternative distances are also explored in Appendix B). However, enforcing similarity across all semantic descriptors may be detrimental, as certain semantic components may be visually present but inaudible, or vice versa. To address this, alignment is applied only to the closest audio and image semantic components. This choice is ablated in Section 4. Using the $L_2$ norm for reconstruction, we obtain $V_A, V_I, U_A,$ and $U_I$ by solving the following optimization problem:

$$\min_{V_A, V_I, U_A, U_I \geq 0} \left[ \|X_A - U_A V_A\|_2^2 + \|X_I - U_I V_I\|_2^2 + \beta_p \min_k \text{CE}(\mathcal{D}_{I^k}, \mathcal{D}_{A^k}) \right], \tag{6}$$

where CE denotes the cross-entropy function. The pseudo-code of **TACO** is given in Appendix C.

**Interpretation** After optimization, the $k^{\text{th}}$ row of $U_I$ represents the segmentation of the $k^{\text{th}}$ factor in $V_I$ (i.e. its $k^{\text{th}}$ column) within the original image. The index $k$ that realizes $\min_k \text{CE}(\mathcal{D}_{I^k}, \mathcal{D}_{A^k})$ corresponds to the dominant semantic component shared between the audio and image modalities, denoted as $k^\star$. Consequently, row $k^\star$ of $U_I$, referred to as $U_I^{k^\star}$, can be interpreted as the segmentation of the image region associated with the sound in the audio signal. Likewise, the segmentation of the spectrogram for this same semantic component is represented by $U_A^{k^\star}$.

**Temporal consistency** To incorporate the temporal dimension in both the audio and visual inputs, if dealing with a video sequence, we introduce a temporal consistency regularization term in the decomposition process. Given a video, we extract $T$ frames at a constant interval, and construct $T$ audio-image pairs, denoted by $\{X_{A_t}\}_{t=1}^T$ and $\{X_{I_t}\}_{t=1}^T$, by extracting the $T$ consecutive non-overlapped audio frames centered around an image frame position. For each frame pair, we perform a decomposition into matrices $V_{A_t}, V_{I_t}, U_{A_t}, U_{I_t}$, introducing a regularization term $\mathcal{R}_t$ to encourage consistency between the primary shared factors, $V_{I_t}^{k^\star}$ and $V_{A_t}^{k^\star}$, across consecutive frames. Formally, we add the following $\mathcal{R}_t$ regularization term to the optimization objective of Eq. equation 6:

$$\mathcal{R}_t = -\beta_{temp} \sum_{t=1}^T \cos(V_{I_t}^{k^\star}, V_{I_{t+1}}^{k^\star}) + \cos(V_{A_t}^{k^\star}, V_{A_{t+1}}^{k^\star}), \tag{7}$$

where $\beta_{temp} \geq 0$ controls the temporal regularization.

## 3.3 Improved Segmentation with an Open Vocabulary Segmenter

Our soft co-NMF approach generates a segmentation mask for objects producing sound in the audio input through the $k^\star$-th row of $U_I$. However, the segmentation accuracy is limited, as the CLIP features in the

visual input $X_I$ were trained only for image-level semantic alignment, not for precise segmentation. To enhance segmentation quality, we propose to integrate a pre-trained open-vocabulary segmentation model. To maintain compatibility with our NMF framework and keep the approach zero-shot, we select a model based on CLIP that preserves alignment with the text encoder without additional fine-tuning of the CLIP encoder. Here, we use the FC-CLIP model (Yu et al., 2023b), which segments images using CLIP's image encoder via text prompts encoded using the CLIP text encoder. A valuable feature of FC-CLIP is its shared CLIP embedding space for both image and text inputs, which allows it to be prompted by vectors from the image space. Leveraging this, we prompt the FC-CLIP decoder with factors identified by our NMF framework. More precisely, in our soft co-NMF decomposition, the rows of matrix $V_I$ represent the factors in the image space. Therefore, we can use them to prompt FC-CLIP, treating each factor in $V_I$ as a segmentation class. Specifically, $V_I^{k^\star}$ describes the shared "sounding" factor between audio and visual inputs. Since $V_I^{k^\star}$ resides in the CLIP space, it can also be compared to class name embeddings, allowing us to infer the segmented object's class names and achieve *sound prompted image semantic segmentation*. The complete pipeline is depicted in Figure 2. Note that, incorporating FC-CLIP is computationally efficient, as the CLIP representation is computed only once before the soft co-NMF decomposition, with the lightweight FC-CLIP decoder adding minimal overhead (there is ∼200M parameters for the CLIP encoder and ∼20M for the FC-CLIP decoder).

## 4 Results and Discussion

**Implementation details.** To extract the modality-specific representations we utilize the MSCLAP 2023 (Elizalde et al., 2023) (∼80M parameters) audio backbone and the OpenCLIP (Cherti et al., 2023) ConvNeXt-Large CLIP trained on LAION 2B (Schuhmann et al., 2022) (∼200M parameters) as they exhibit higher performance and generalization than the aligned audio-visual models in audio and image only tasks (Elizalde et al., 2023). We fix the decomposition coefficients to $\beta_p = 125$, and $K = 8$, and we ablate those choices in Appendix B. All the matrices $U_A, V_A, U_I$ and $V_I$ are initialized randomly from Gaussians. We set the image and audio frame rate to 1 and use $\beta_{temp} = 1$ for the temporal consistency constraint. Since it targets single-source segmentation, we set $\beta_{temp} = 0$ for the multi-source task. Those hyperparameters were selected based on preliminary experiments conducted on the AVSBench validation set. The optimization problem is solved using gradient descent in 1800 steps with a learning rate of 0.25. Note that, this optimization induces minimal computational overhead as detailed in the Appendix 4.5). For semantic segmentation, we use the class names provided with the dataset as the word bank. For non-semantic segmentation, no predefined class set is needed, we therefore used a general class list: the set of 527 AudioSet(Gemmeke et al., 2017) tags. We ablate this choice in Appendix B. Since the $U_I$ and $U_A$ matrices are randomly initialized, we ran each experiment three times and report the average score as well as the variance.

**Datasets and Metrics.** We evaluate our method on five datasets: three for sound-prompted segmentation and two for sound-prompted semantic segmentation. For sound-prompted segmentation, we use: *AVS-Bench* (Zhou et al., 2022), with binary segmentation maps for audio-visually related pixels, divided into single-source (S4, 5K samples) and multi-source (MS3, 400 samples) datasets. We also use *ADE20k sound-prompted* (Hamilton et al., 2024), which pairs ADE20k segmentation masks and images(Zhou et al., 2019) with audio from VGGSound (Chen et al., 2020). For sound-prompted semantic segmentation, we created *ADE Sound Prompted Semantic*: a semantic version of the ADE20k sound-prompted dataset, by assigning class labels to segmentation masks. We also conduct experiments on *AVSS* (Zhou et al., 2025), (∼11K samples) comprising single and multi-source sounding objects. Regarding metrics, we adopt standard segmentation metrics for evaluation: **mean** Intersection over Union (mIoU) and F-score for the semantic segmentation tasks, and mean Average Precision (mAP) as well as **mask** Intersection over Union (mask-IoU) and F-score for the binary segmentation tasks. Each experiment involving NMF decomposition was conducted three times, with the results reported as the average and standard deviation of the scores. **Baseline.** In the absence of prior work on unsupervised sound-prompted *semantic* segmentation, and to have a fair competitor that relies on the same models, we consider a straightforward yet effective baseline approach: Audio-prompted-FC-CLIP (A-FC-CLIP). This method takes an audio input and its corresponding image, utilizing CLAP to determine the most likely class from the dataset's class pool. FC-CLIP is then prompted to segment this class.

To investigate the resilience of CLIP representations to clamping negative values, using the baseline setup, we prompted FC-CLIP with only the non-negative part of the CLIP text encoding and evaluated its performance.

Results in Appendix A confirm the validity of clamping, supporting the use of NMF-based methods to leverage their interpretability.

In the following subsection, we compare our approach with other *unsupervised* methods tackling the task of sound-prompted segmentation. Most of the methods rely on fine-tuning (even if done unsupervisedly), which significantly modifies the original representations of pre-trained models. In contrast, our method is training-free and fully preserves the generalization ability of the models it builds upon.

## 4.1 Sound-prompted Segmentation

We now discuss experiments on (non-semantic) Sound-Prompted Segmentation. Table 1 compares our approach with other unsupervised methods on the AVS-Bench, in both single and multi-source setups. For completeness, we also include in the comparison Box-Prompt (Yu et al., 2023a), a method which benefits from SAM, a much richer segmentation model than FC-CLIP, making the comparison not fair. Importantly, this approach also assumes that the number of sources known a priori (one in S4 and two in MS3—information not available in real world scenario), while **TACO** does not rely on this additional information. In contrast, our method is able to extract audiovisual factors and encodes a variable number of sources within the sounding factor. **TACO** demonstrates strong performance relative to unsupervised training-based alternatives, outperforming the best competitor method by ~5 and ~2 points of mIoU in S4 and MS3 tasks. Our method also largely outperforms the A-FC-CLIP baseline that uses the same pre-trained models, underlining the effectiveness of our decomposition approach. Interestingly, using $U_I^{k^\star}$ directly as a segmentation map yields results competitive with state-of-the-art methods, despite our model never being exposed to any audiovisual data (only outperformed by MarginNCE (Park et al., 2023) in terms of F-score). **TACO** even outperforms Box-Prompt, despite the latter's reliance on SAM and its access to prior information regarding the number of sources.

In the multi-source setup, while baseline performance drops substantially due to its limitation in handling only a single concept, our method shows robust performance. We hypothesize that robustness arises because $V_I^{k^\star}$ encodes information about **all** the sounding objects rather than just one, within the sounding factor. The hypothesis is explored qualitatively in Section 4.3.1. Table 2 presents performance results on the recently proposed ADE Sound Prompted (ADE SP) dataset, along with variations of our approach on the same dataset to understand its gains. We compare existing methods with **TACO** variants and our decomposition without FC-CLIP. As in AVSBench, using only $U_I^{k^\star}$ from our decomposition yields strong segmentation, outperforming prior audio-visual models. Notably, unlike methods such as DenseAV (Hamilton et al., 2024), CAVMAE (Gong et al., 2023), DAVENet (Hsu et al., 2019), and ImageBind (Girdhar et al., 2023), which are trained on audio-image pairs, our approach uses only frozen features. By decomposing these features to identify matching factors, we show that they inherently capture the spatial information needed for localization. Predictably, the use of FC-CLIP segmenter substantially refines the segmentation and the scores. Designing a decomposition specifically tailored to be a compatible prompt for FC-CLIP is a core component of our method and a key direction for achieving significantly better results.

Remarkably, **TACO** matches the results of Box-Prompt without requiring SAM or prior knowledge of

| Method | Training-free | Segmenter | #Sources | S4 | | MS3 | |
| --- | --- | --- | --- | --- | --- | --- | --- |
| | | | | mask-IoU ↑ | F-Score ↑ | mask-IoU ↑ | F-Score ↑ |
| SLAVC (Mo & Morgado, 2022b) | ✗ | ✗ | ✗ | 28.10 | 34.60 | 24.37 | 25.56 |
| MarginNCE (Park et al., 2023) | ✗ | ✗ | ✗ | 33.27 | 45.33 | 27.31 | 31.56 |
| FNAC (Senocak et al., 2023) | ✗ | ✗ | ✗ | 27.15 | 31.40 | 21.98 | 22.50 |
| Alignment (Sun et al., 2023) | ✗ | ✗ | ✗ | 29.60 | 35.90 | - | - |
| TACO w/o segmenter ($U_I^{k^\star}$) | ✓ | ✗ | ✗ | 29.68±0.16 | 41.91±0.09 | 25.88±0.91 | 30.72±0.95 |
| Box-Prompt (Yu et al., 2023a) | ✓ | SAM | ✓ | 51.2 | 61.5 | 41.8 | **47.8** |
| ACL-SSL (Park et al., 2024) | ✗ | CLIPSeg | ✗ | 59.76 | 69.03 | 41.08 | 46.67 |
| A-FC-CLIP | ✓ | FC-CLIP | ✗ | 51.52 | 57.25 | 33.67 | 35.87 |
| TACO | ✓ | FC-CLIP | ✗ | **64.04± 0.25\*** | **71.50±0.20\*** | **43.15 ±0.91** | 47.5±0.95 |

Table 1: Quantitative results of sound-prompted segmentation on the AVS-Bench test sets. Best is in **bold** and second best underlined. The column #Sources indicates whether the number of sources is known. The * indicates a p-value lower than 0.01 when comparing to the best performing model.

the source count. It even outperforms Box-Prompt in terms of mask-IoU. This demonstrates the inherent robustness of our method when handling multiple sources. To gain further insight, we now examine variants of **TACO**. In the *TACO w/o component* variant, using $V_I^k$ and $V_A^k$ instead of the soft mask representations as semantic components during the optimization process (i.e., using $\mathcal{D}_I^k = V_I^k$ and $\mathcal{D}_A^k = V_A^k$) leads to degraded performance. In *TACO No-min*, we replace the min operator in equation 6 with an average, penalizing all concept distances instead of just the closest. This results in a noticeable drop in performance. The last variant, *TACO w/o penalty*, shows the results using $\beta_p = 0$ (i.e., without penalization), which is equivalent to performing two independent decompositions and choosing $k^\star = \arg\min_k CE(\mathcal{D}_I^k, \mathcal{D}_A^k)$ ;resulting in a big performance drop.

## 4.2 Sound-Prompted Semantic Segmentation

Sound-Prompted Semantic Segmentation differs from Sound-Prompted Segmentation seen in the previous section as it requires the estimation of class labels corresponding to the mask. In **TACO**, classifying the audio prompt $V_I^{k^\star}$ allows us to obtain the label of the segment. We run experiments on the AVSS and ADE20k SP Semantic datasets. As shown in Table 3, our method greatly outperforms the baseline on the two datasets as it can take advantage of both the image and the audio decompositions (+8 and 15 points of mIoU in AVSS and ADE SP Semantic, respectively).
On the ADE SP Semantic, which contains a single class per image, the method performs well. However, on the AVSS dataset that contains samples with multiple classes, the performances drop. We attribute this to the class decoding process: since we estimate the class label as the closest one to the value of $V_I^{k^\star}$, we consider the full segmentation as a single class. Nonetheless, the class-agnostic segmentation masks are accurate as the decomposition encodes all the concepts in a single vector (shown in Appendix F).

## 4.3 Qualitative Analysis

This qualitative analysis is structured in two parts: we first assess **TACO**'s performance across diverse scenarios, and then deepen our understanding through a set of ablation studies.

### 4.3.1 Qualitative Analysis of TACO

We begin this qualitative analysis by presenting in Figure 4 input images along with their associated $V_I^{k^\star}$ reshaped as a matrix, the segmentation from **TACO**, and an illustration of the associated audio. These examples highlight both the segmentation quality of **TACO** and the method's interpretability. Specifically, $U_I^{k^\star}$ reveals the tokens attended by the decomposition, from which $V_I^{k^\star}$ serves as the prompt for FC-CLIP. The first row shows a challenging example from the S4 dataset, where the sound source is nearly invisible, yet both **TACO** and $V_I^{k^\star}$ successfully segment it. While recent works (Juanola et al., 2025) underline how audio-visual segmenter models fail to deal with *negative audio* (i.e. audio depicting an event not present in the image), the second row illustrates the robustness of our method on that particular problem. As there is no common concept between the audio and image (given that the sound track of this example features

| | Segmenter | Training-free | m-IOU↑ | mAP↑ |
|---|---|---|---|---|
| DAVENet (Hsu et al., 2019) | - | ✗ | 17.0 | 16.8 |
| CAVMAE (Gong et al., 2023) | - | ✗ | 20.5 | 26.0 |
| ImageBind (Girdhar et al., 2023) | - | ✗ | 18.3 | 18.1 |
| DenseAV (Hamilton et al., 2024) | - | ✗ | 25.5 | 32.4 |
| TACO w/o segmenter ($U_I^{k^\star}$) | - | ✓ | 27.74±0.48 | 35.75±0.33 |
| A-FC-CLIP | FC-CLIP | ✓ | 45.31 | 50.36 |
| TACO    No-min | | ✓ | 29.93±1.95 | 34.59±3.17 |
|    w/o component | FC-CLIP | ✓ | 22.70±0.74 | 23.68±2.03 |
|    w/o penalty | | ✓ | 26.75±0.14 | 30.17±0.12 |
|    Proposed | | ✓ | **51.57**±0.86* | **60.01**±1.38* |

Table 2: Quantitative results of sound-prompted segmentation on the ADE SP dataset. The * indicates a p-value lower than 0.01 when comparing to the best performing model.

| Method | AVSS | | ADE SP Semantic | |
|---|---|---|---|---|
| | mIoU ↑ | F-Score ↑ | mIoU ↑ | F-Score ↑ |
| A-FC-CLIP | 11.95 | 13.56 | 20.91 | 24.95 |
| TACO | **20.50** ±0.10* | **23.02** ±0.09* | **37.70**±1.61* | **44.39**±1.92* |

Table 3: Quantitative results of sound-prompted semantic segmentation on the AVSS and ADE SP Semantic datasets. The * indicates a p-value lower than 0.01 when comparing to the best performing model.

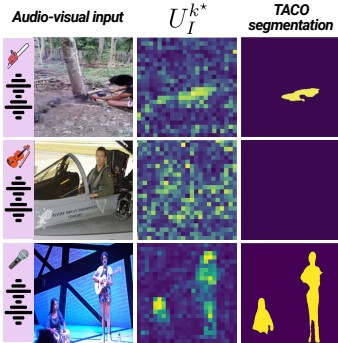

Figure 4: Qualitative segmentation examples from ADE SP and S4 datasets: **TACO** performs well even in challenging cases.

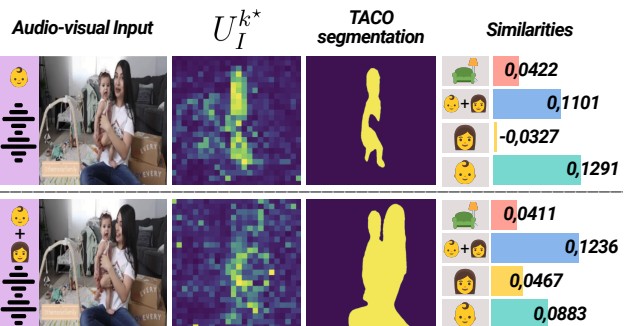

Figure 5: Multiple source segmentation examples. As the sources change during the video, **TACO**'s segmentation changes as well.

music instead of the original audio recording of the scene), the estimated $V_I^{k^\star}$ contains only noise, therefore FC-CLIP does not find it in the image, hence the final segmentation is empty, showing the resilience of **TACO** to those negative audios. Finally, the last row illustrates the case of multiple potential sources in the image. Even though the image features a Guitar, a Djembe, and singers, the model correctly segments only the people as only singing voice is heard on this excerpt.

Figure 5 shows a segmentation example from the multi-source MS3 dataset. To gain insight into the model's behavior, we manually selected words describing sounding entities in the image and examined the similarity $V_I^{k^\star}$ with them. In the first row, the sound heard is a baby babbling (👶 in the first column), while the image contains both the baby, a woman, and a sofa. As expected, the closest word to $V_I^{k^\star}$ is "Baby babbling" (see 👶 in the last column). In the second row, both the women and the baby are making sounds (👶+👩 in the first column). While one might expect these two sounds to be encoded in two different factors, they are both encoded in the same sounding factor $V_I^{k^\star}$, as its similarity is maximal with "A baby babbling and a woman talking" (👶+👩 in the last column), which is coherent with the segmentation. This observation reinforces our hypothesis made in section 4.1: the decomposition encodes not just a common concept between audio and image features, but **all** the sounding objects, explaining its performance in the multi-source task.

### 4.3.2 Qualitative Ablation studies

In this section, we evaluate the impact of two components of the method: the penalty loss and the temporal consistency loss. First, figure 6 shows qualitative examples obtained by **TACO** on the S4 dataset, with and without the penalty loss. Not penalizing the decomposition to be performed jointly is essentially equivalent to running two independent NMFs (one on the image and one on the audio features), and selecting the sounding factor as the one that minimizes the distance between semantic descriptors ($\text{CE}(\mathcal{D}_{I^k}, \mathcal{D}_{A^k})$). However, this strategy does not take into account the mutual dependence between the visual and auditory sources, and therefore cannot ensure that the selected components represent the same underlying object, and base the decomposition on sounding/non-sounding factors. While it may perform well when a single dominant concept is clearly present in both modalities (see the bus example in the figure), it struggles in scenes where either the image or the audio contains multiple salient elements, resulting in incorrect or fragmented localization. Second, figure 7 illustrates the impact of temporal consistency on the S4 dataset. Even when the visual

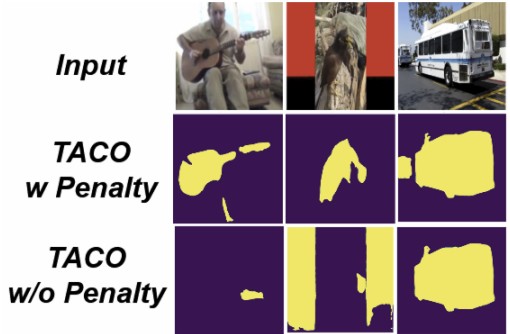

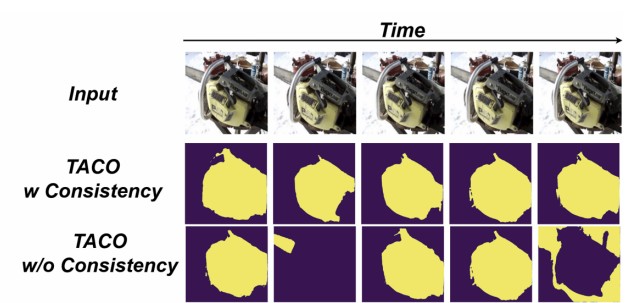

Figure 6: Qualitative visualization of the impact of the penalty loss on the segmentation in the S4 dataset.

Figure 7: Qualitative comparison of TACO segmentation with and without temporal consistency on the S4 dataset

content does not change significantly over time, the audio may vary, making segmentation more challenging at different timesteps. Temporal consistency helps in such cases by encouraging the model to maintain the same sounding factor across timestamps.

## 4.4 Word bank Analysis

Figure 8 shows the mIoU of **TACO** on the S4 dataset when varying the proportion of AudioSet tags used in the word bank.

Interestingly, performance remains stable until half of the tags are removed. It then drops when using only $\sim 20\%$ of the tags, and even more significantly when fewer tags are used. This behavior underlines that AudioSet tags are diverse enough to express a wide range of concepts, and that removing a large portion of them (half) does not significantly impact the model.

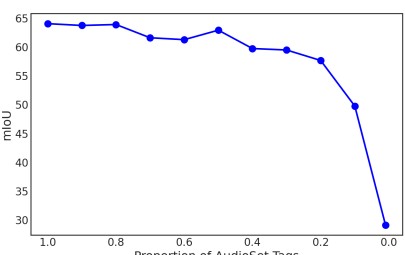

Figure 8: mIoU of **TACO** on S4, with varying number of tags in the word bank.

In order to vary the content of the word bank, and not only its size, we prompted GPT-5.2 to generate a large list of sounding objects (407), which we used as the word bank in **TACO**. Table 4 compares the performance of **TACO** using the AudioSet word bank and the generated one. The difference in performance is negligible (less than 0.2 in mask-IoU and 0.35 in F-score). This not only demonstrates the robustness of our method to variations in the word bank, but also shows that constructing this list is not a bottleneck in **TACO**.

| Word Bank | mask-IoU ↑ | F-Score ↑ |
|---|---|---|
| AudioSet tags | **64.05** ±0.25 | **71.50** ±0.20 |
| GPT5.2 generated | 63.83 ±0.29 | 71.18 ±0.29 |

Table 4: Ablation on S4 dataset: choice of general word bank.

## 4.5 Computational overhead

The proposed NMF-based decomposition introduces additional computational costs compared to the baseline (consisting in prompting FC-CLIP with the predicted class from CLAP) as it requires optimization for each batch/sample. Figure 9 shows the performance (mAP) of **TACO** depending on the number of iterations in the optimization process as well as the associated relative computing time for a complete inference on the ADE SP test set. The performance plateaus after $\sim 1000$ iterations, which takes $\sim 20\%$ additional time compared to the baseline on a single Nvidia A100 80G. Additionally, the amount of memory required to

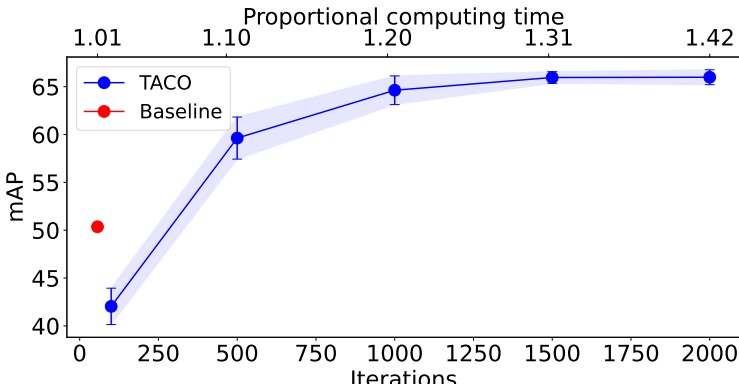

Figure 9: Proportion of additional computing time induced by the decomposition. The decomposition allows an improvement of $\sim 12$ points with only $\sim 20\%$ of additional compute time compared to the baseline (A-FC-CLIP).

decompose an audiovisual input is $\sim 425,\mathrm{MB}$ (576 image tokens for an image with resolution $764 \times 764$, and 64 audio tokens for a 10-second audio clip).

## 5 Conclusion and limitations

We introduced **TACO**, a method for training-free sound-prompted visual segmentation, effectively incorporating an open vocabulary image segmentation model, namely FC-CLIP, by prompting it with factors extracted through a semantically motivated co-factorization of audio-visual deep embedding representations. Our method provides a framework for unsupervised audio-visual segmentation, enabling the extraction and visualization of sounding concepts shared across audio spectrograms and image regions, making it inherently interpretable. **TACO** outperforms concurrent approaches, achieving state-of-the-art results. However, although the Sem CO-NMF incurs lower costs compared to pretrained models, it still introduces additional computational overhead during inference, which may pose challenges in resource-constrained settings. Finally, while **TACO** can support semantic segmentation, the decoding of multiple labels in the same image remains challenging and should be improved in further work.

## 6 Acknowledgement

This work was supported by the Audible project, funded by French BPI. This work was performed using AI resources from GENCI-IDRIS (Grant 2025-AD011014885R1)

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

The appendix is organized as follows: the first part presents an analysis of the clamping resilience as well as additional ablations and hyperparameter variations. The second part presents a detailed pseudo-code of the method, followed by an analysis of the encoding costs induced by the pretrained models. The last parts discuss the metrics used and show additional qualitative visualizations and analysis of segmentation from **TACO**.

## A  Analysis of clamping resilience

To assess the resilience of CLIP representations to clamping negative values, we evaluate FC-CLIP using only the nonnegative portion (obtained via ReLU) of the CLIP text encoding, a variant referred to as $A^+$-FC-CLIP. Table 5 shows the performance degradation across all datasets. Except for the ADE SP dataset ($\sim$3.5 points drop), the decrease never exceeds 2 points. This experiment validates the use of clamping, enabling the application of NMF-based methods so as to benefit from their interpretability.

|              | ADE SP   | S4       | MS3    | AVSS     | ADE SP S. |
|--------------|----------|----------|--------|----------|-----------|
| A-FC-CLIP    | **45.31**| 51.52    | 33.67  | **11.95**| **24.95** |
| $A^+$-FC-CLIP| 41.89    | **51.63**| 31.67  | 11.82    | 23.73     |

Table 5: Impact of clamping on the representation before feeding FC-CLIP. We report mIoU on five datasets. *ADE SP S.* corresponds to the ADE SP Semantic segmentation dataset. mask-IoU is reported for S4 and S3 and mIoU for all the other datasets.

## B  Analysis of design choices

**Hyperparameter analysis.**  Table 6 shows the performance of our method on the S4 dataset using different numbers of components in the decomposition (values of K). Results are stable across multiple values, showing the resilience of the method with respect to the number of factors.

Table 7 shows the performance of **TACO** (on the S4 dataset) with respect to the hyperparameter $\beta_p$. The performance remaining stable across multiple values of the hyperparameter indicates the robustness of the method. TACO-KL shows the performance of **TACO** when using the KL-divergence instead of the cross entropy to measure dissimilarity between the semantic descriptors. Using KL-divergence instead of cross entropy affects significantly the performance of **TACO**.

Additionally, we ablate the temporal consistency component by setting $\beta_{temp} = 0$. This significantly degrades performance, highlighting the effectiveness of that regularization.  Table 8 presents the mask-IoU scores on

| S4  |            |            |
|-----|------------|------------|
| K   | mask-IoU ↑ | F-Score ↑  |
| 6   | 61.74      | 68.87      |
| 8   | **64.04**  | 71.50      |
| 10  | 63.92      | **71.54**  |
| 12  | 63.78      | 71.28      |

Table 6: Variation of the number of factors

| S4 | | | | |
|----|----|----|----|----|
| Method | $\beta_p$ | $\beta_{temp}$ | mask-IoU ↑ | F-Score ↑ |
| TACO | 100 | 1 | 63.45 | 70.57 |
|      | 125 | 1 | **64.04** | 71.50 |
|      | 150 | 1 | 63.90 | **71.52** |
| TACO | 125 | 0 | 53.47 | 59.93 |
| TACO-KL | 125 | 1 | 58.74 | 65.69 |
| TACO Words/2 | 125 | 1 | 62.91 | 70.16 |
| TACO Words/4 | 125 | 1 | 55.50 | 62.10 |

Table 7: Variation of the design choices

the three used sound-prompted segmentation datasets using either a general class list or one specific to the dataset (i.e. the list of all the classes of the dataset). Using a specific class list overall improves both **TACO** results (+1 point on S4, 3.5 points on ADE SP and 2 points in MS3). Interestingly, the segmentation using $U_i^{k^\star}$ does not seems to be improved a lot by the specific word bank (+1 point on S4 and MS3, and -5 points on ADE SP).

**Formulation choices.** Because multiple-sound-source (MS3) scenarios are more complex and challenging, they provide a complementary test of the contribution of each component of our method. As such, Table 9 reports the results of **TACO** with and without our penalty function. We obtained results in line with what we observe in the single-sound-source case (see Table 2): without penalty, the performance is almost halved, underlining the effectiveness of our proposed mechanism. In order to verify whether the sounding factor encodes all sounding sources, even in the multi-source setting, we modified the formulation of the penalty loss and penalized the *bottom-2* values instead of taking only the minimum, i.e., we forced two factors to be shared between the image and the audio. Line *bottom-2* of Table 9 shows the performance of **TACO** using both the *bottom-2* formulations of the penalty function. Using the original min formulation yields better results, indicating that the decomposition more effectively encodes all sounding factors into a single concept than when two are forced.

| Method | mask-IoU ↑ | F-Score ↑ |
|---|---|---|
| TACO | **43.15** ±0.91 | **47.5** ±0.95 |
| TACO w/o penalty | 23.52 ±0.54 | 24.23 ±0.65 |
| TACO bottom-2 | 41.12 ±0.90 | 45.17 ±1.14 |

Table 9: Ablation on MS3 dataset: impact of penalty function.

## C   TACO pseudo-code

The following pseudo-code details the exact computations performed by **TACO**. Note that this is the pseudo code for the case of a static frame, in the case of multiple frames (like a video) Eq equation 7 should be added to $\mathcal{L}$.

The visualization of the decomposition is given in Figure 10. The input matrices are decomposed in 2 matrices: $U_A$ and $V_A$ and $U_I$ and $V_I$. The kth column of $U_A$ and $U_I$ that minimizes the cross entropy between the semantic descriptors is used to force the unify the 2 decompositions.

## D   Encoder MACS analysis

The table 10 shows the cost (measured in MACs) of the image and audio encoder used in our method and concurrent ones. While our image encoder is larger, the efficiency of the CLAP audio encoder (HTSAT- Swin Transformer) significantly reduces the overall computation. As a result, our total encoding cost is lower than that of Hamilton et al. (2024) and Park et al. (2024).

## E   Metrics discussion

Throughout this work, we reported mask-IoU as one of the primary metrics for sound-prompted segmentation on the AVSBench dataset, instead of mean-IoU. While mean-IoU is commonly used in segmentation tasks, it may not be the most appropriate metric in the context of binary segmentation. This is because mean-IoU averages the IoU of both the foreground and the background $(\text{IoU(foreground)} + \text{IoU(background)})/2$, potentially overemphasizing background accuracy, which is often less critical in binary tasks. In contrast,

| | Word Bank | S4 | MS3 | ADE SP |
|---|---|---|---|---|
| $U_i^{k^\star}$ | General | 27.78±0.06 | 25.88±0.91 | 32.42±0.74 |
| | Specific | 28.76±0.06 | 26.57±0.17 | 27.74±0.48 |
| TACO | Specific | 64.73±0.16 | 45.15±0.80 | 54.05±1.40 |
| | General | 64.04±0.25 | 43.15±0.91 | 51.57±0.86 |

Table 8: Comparison of the two word banks on sound-prompted segmentation datasets. mask-IoU is reported for S4 and MS3, while mIoU is reported for ADE SP.

---

**Algorithm 1** TACO algorithm

---

1: **Input:** Audio and image features $(X_A, X_I)$, number of iterations $n_{\text{ite}}$, semantic anchors $\{b_I^j\}_{j=1}^J$, $\{b_A^j\}_{j=1}^J$, learning rate $\eta$, parameter $\beta_p$, FCCLIP model $s(.)$

2: **Initialize:** $U_A$, $U_I$, $V_A$, $V_I \sim \mathcal{N}(0, \Sigma)$

3: **for** iteration $t = 1$ to $n_{\text{ite}}$ **do**

4:     Compute semantic components: $\mathcal{C}_I^k = \text{avg}(X_I \odot U_I^k)$, $\mathcal{C}_A^k = \text{avg}(X_A \odot U_A^k)$

5:     Compute semantic descriptors: $\mathcal{D}_{I^k} = \begin{pmatrix} \cos(\mathcal{C}_I^k, b_I^1) \\ \vdots \\ \cos(\mathcal{C}_I^k, b_I^J) \end{pmatrix}$, $\mathcal{D}_{A^k} = \begin{pmatrix} \cos(\mathcal{C}_A^k, b_A^1) \\ \vdots \\ \cos(\mathcal{C}_A^k, b_A^J) \end{pmatrix}$

6:     Compute total cost:
$$\mathcal{L} = \|X_A - U_A V_A\|_2^2 + \|X_I - U_I V_I\|_2^2$$

7: $+ \beta_p \min_k \text{CE}(\mathcal{D}_{I^k}, \mathcal{D}_{A^k})$

8:     Update parameters:

9:         $U_A \leftarrow U_A - \eta \frac{\partial \mathcal{L}}{\partial U_A}$

10:         $U_I \leftarrow U_I - \eta \frac{\partial \mathcal{L}}{\partial U_I}$

11:         $V_A \leftarrow V_A - \eta \frac{\partial \mathcal{L}}{\partial V_A}$

12:         $V_I \leftarrow V_I - \eta \frac{\partial \mathcal{L}}{\partial V_I}$

13: **end for**

14: $k^\star = \arg\min_k \text{CE}(\mathcal{D}_{I^k}, \mathcal{D}_{A^k})$

15: segmentations $\leftarrow s(V_{I_t}, X_I)$

16: **Return:** segmentation$[k^\star]$

---

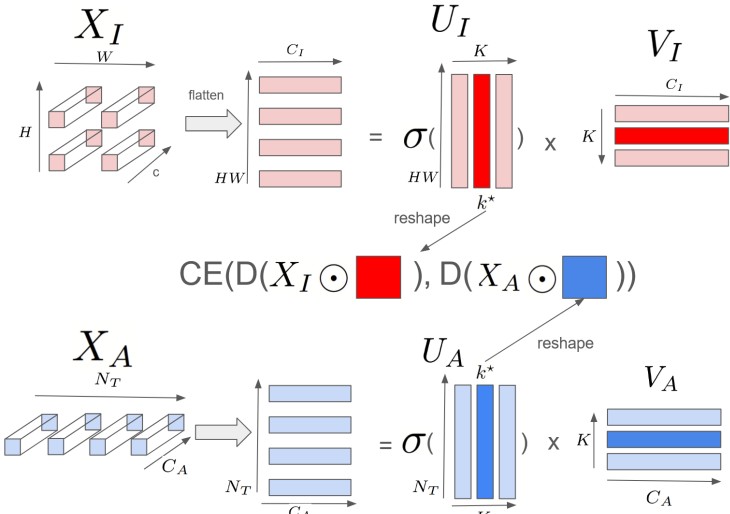

Figure 10: Illustration of the decomposition

| Method | Image Encoder | Audio Encoder | Total |
|---|---|---|---|
| Hamilton et al. (2024) | $1.69 \times 10^{10}$ | $6.72 \times 10^{10}$ | $8.41 \times 10^{10}$ |
| Park et al. (2024) | $1.69 \times 10^{10}$ | $4.48 \times 10^{10}$ | $6.17 \times 10^{10}$ |
| Ours | $3.44 \times 10^{10}$ | $6.31 \times 10^{9}$ | $4.07 \times 10^{10}$ |

Table 10: Comparison of model MACs for image and audio encoders.

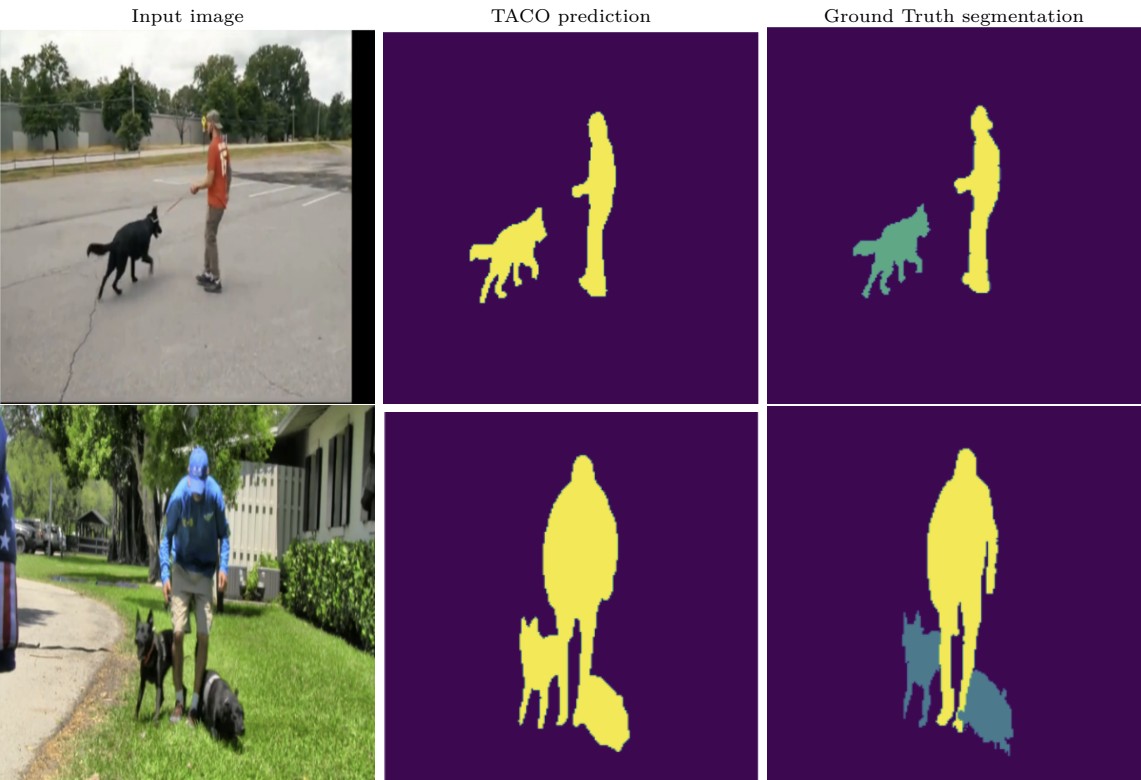

Figure 11: Semantic segmentation examples from AVSS.

mask-IoU focuses solely on the IoU of the foreground, making it a more relevant and precise measure of segmentation quality in scenarios where the foreground object is the primary area of interest. However, we noticed that several works report mean-IoU in their papers, while in practice they measure mask-IoU. This discrepancy appears to stem from unintentionally replicating an error in the original repository of dataset AVSBench, which leads to potentially unfair comparisons between works. This discrepancy poses some challenges when comparing results, particularly for works without available code as the exact metric used cannot always be confirmed. Consequently, we were unable to compare our results with works for which the code was not available (like Bhosale et al.), as there was no assurance regarding the metrics used.

# F   Semantic segmentation analysis

Figure 11 presents qualitative evidence supporting the hypothesis mentioned in Section 4.2: the semantic segmentation performance is hindered by the class inference process. Specifically, in a scenario with two sound sources, while both sources may be well segmented, each being encoded within the sounding concept through decomposition, the use of a single factor to infer the class leads to both sources being assigned the same class. This significantly penalizes the segmentation score.

## G    Qualitative results of single and multiple sources, sound-prompted segmentation

Figure 12 presents randomly sampled segmentation examples from the ADE Sound prompted dataset, as well as their associated activation matrices $U_I^{k^\star}$, which enlightens the regions of the image that is used to extract the prompt to the segmenter model.
Figure 13 presents the equivalent analysis applied to the multi-sources dataset MS3.

## H    Typical failure cases

Figure 14 shows typical failure cases of **TACO**. The first row depicts a case where the decomposition correctly identifies that the concept to segment is a piano (as shown by $U_I^{k^\star}$), but the segmenter model fails to clearly segment it, resulting in inaccurate segmentation. The second row showcases an example where the decomposition fails to extract a clear concept, as it is unable to encode both audio concepts effectively, yielding partial and inaccurate segmentation. Finally, the third row highlights a special failure case where the decomposition successfully encodes one concept but neglects the second. We hypothesize that this issue arises from limitations in the performance of the audio model (CLAP audio encoder).

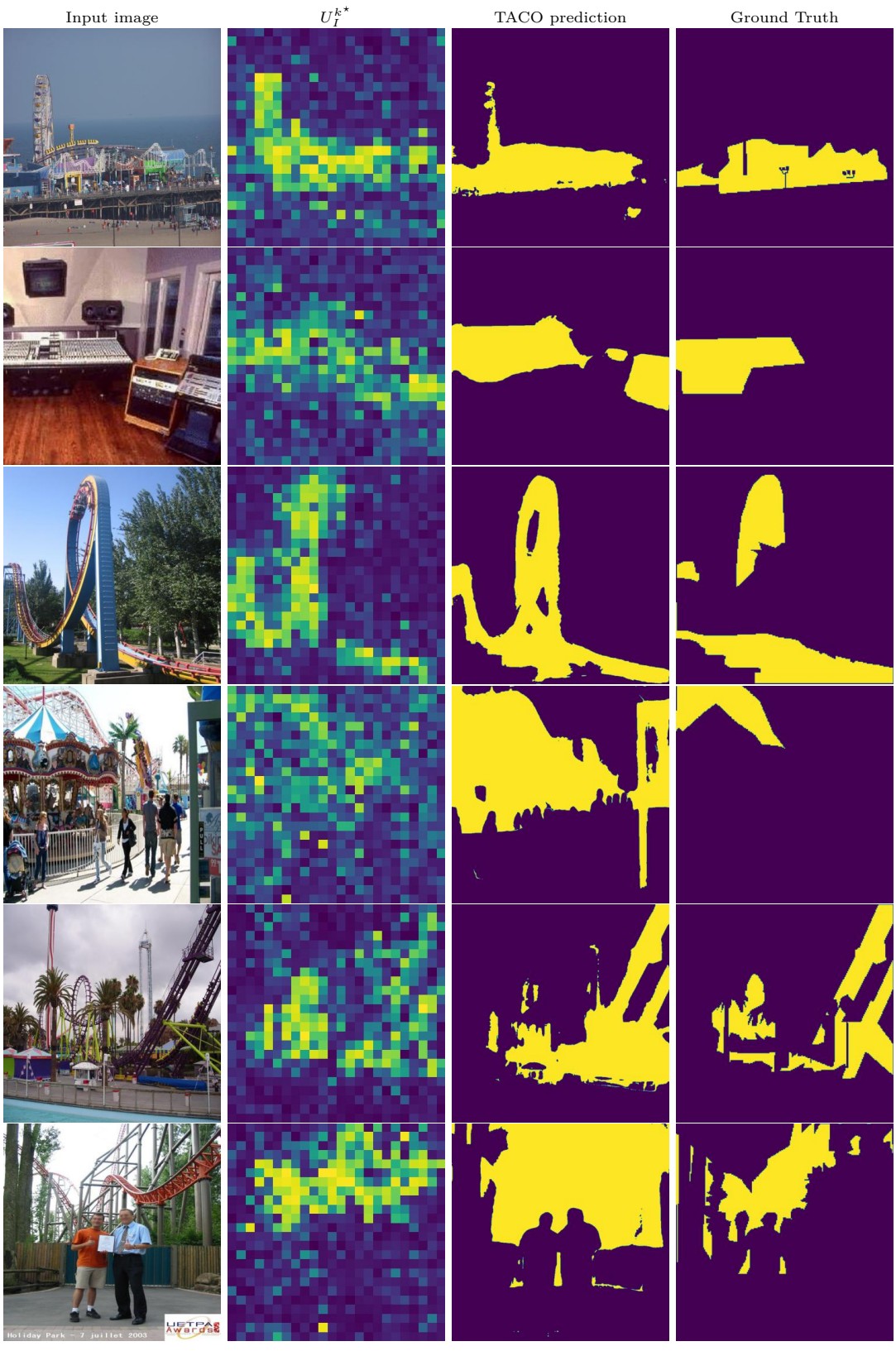

Figure 12: Sound-prompted segmentation examples from ADE SP.

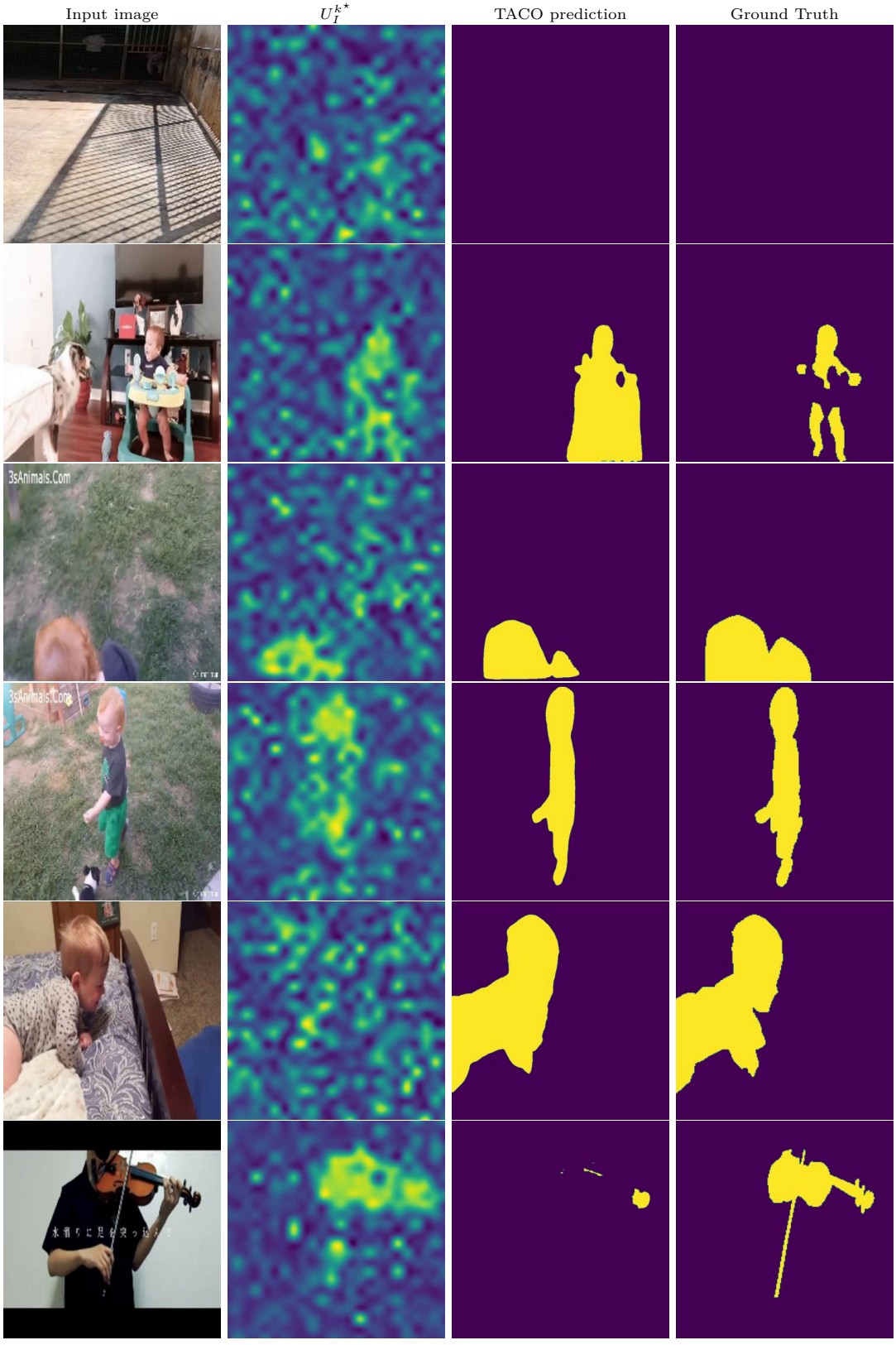

Figure 13: Sound prompted-segmentation examples from MS3.

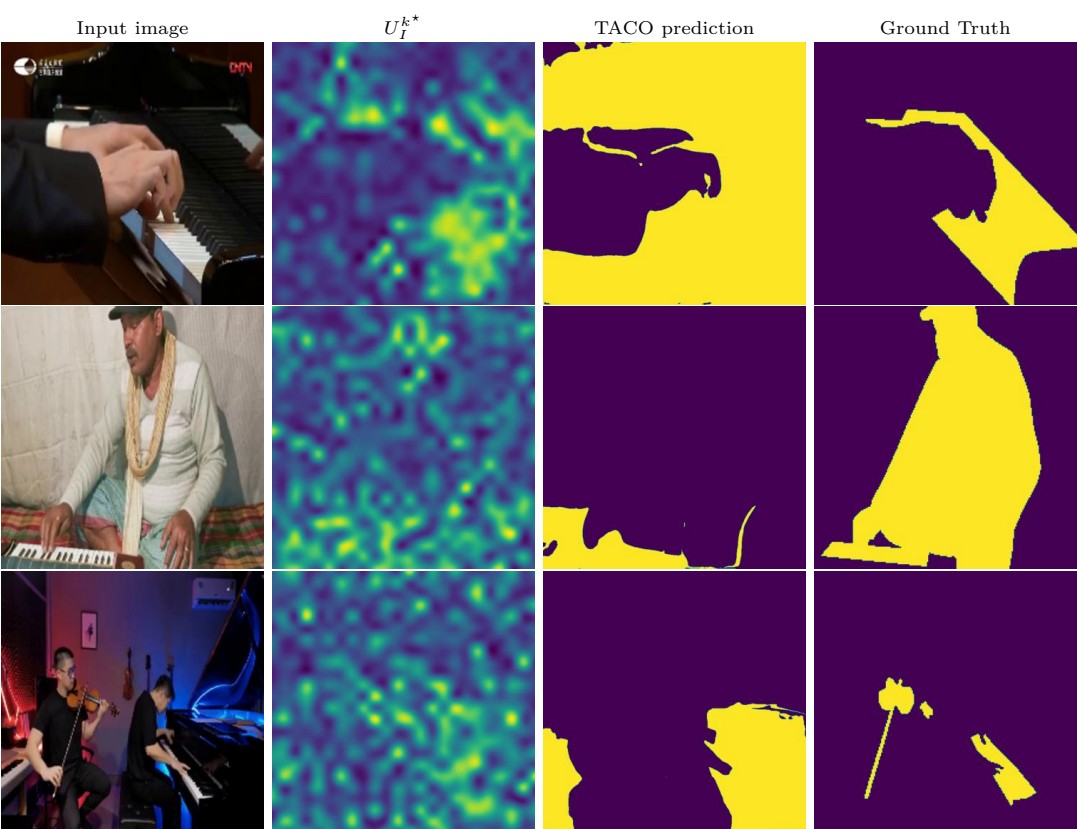

Figure 14: Examples of typical failure cases on MS3.

