# OpenReview forum: "TACO: Training-free Sound Prompted Segmentation via Semantically Constrained Audio-visual CO-factorization"
_TMLR — Accepted by TMLR_

### Review · Reviewer_MPy6 · 2025-12-14

**Summary Of Contributions:**

This paper proposes TACO, a training-free framework for sound-prompted image segmentation, where the goal is to segment objects in a static image that correspond to a given audio signal. The method leverages frozen, large-scale pre-trained models—CLIP for vision and CLAP for audio—and introduces a semantically constrained co-nonnegative matrix factorization (co-NMF) to align audio and visual representations without any additional model training.
The key idea is to jointly decompose audio and image features into a shared set of low-rank semantic factors, which are further regularized by text-based semantic anchors derived from CLIP’s language space. These factors provide coarse spatial localization, which is then refined using an open-vocabulary segmentation model (FC-CLIP) to obtain high-quality masks. The approach is fully unsupervised, doesn't rely on paired video supervision, and operates entirely at inference time.
Key strengths:
1. Introduces a novel, principled training-free formulation for sound-prompted segmentation.
2. Effectively reuses powerful pre-trained multimodal models without fine-tuning.
3. The co-NMF formulation yields interpretable semantic factors and spatial activations.
4. Demonstrates strong empirical performance on multiple benchmarks, outperforming prior unsupervised methods by a significant margin.

**Audience:**

Yes

**Audience Explanation:**

The paper addresses a timely and relevant problem at the intersection of multimodal learning, weakly supervised segmentation, and interpretability. Satisfying the audience of TMLR for machine learning.
The training-free nature of TACO is especially appealing in scenarios where labeled data or large-scale retraining is impractical. The use of co-NMF as a semantic alignment tool provides methodological insights, and contributes both a practical system and a conceptual framework for inference-time multimodal alignment.

**Broader Impact Concerns:**

1. Inference efficiency and scaling behavior:
Although the method is training-free, its practical feasibility for real-world industrial deployment remains to be further evaluated. In particular, additional analysis is needed for large-scale and distributed settings, such as inference-time performance under billion-scale data regimes or high-QPS (queries per second) scenarios.
2. Bias and representational limitations:
The method relies on large pre-trained models and benchmark datasets, with evaluations conducted on relatively homogeneous data types. It would be valuable to extend the experiments to a more diverse set of datasets, as well as to compare performance across different types of pre-trained models (e.g., Whisper), to better assess generalization and representational biases.
3. Robustness to noisy or adversarial audio:
The system may be sensitive to noisy, misleading, or adversarial audio signals, which could lead to incorrect segmentation outcomes in real-world applications. Evaluating robustness under such conditions would strengthen the practical relevance of the method.

**Claims And Evidence:**

Yes

**Claims Explanation:**

The paper provides a clear methodological description, including the motivation for using co-NMF, the semantic anchoring mechanism, and the integration with an open-vocabulary segmentation model. The experimental evaluation is extensive, covering several widely used benchmarks for sound-prompted or audio-visual segmentation (e.g., AVS-Bench S4/MS3, AVSS, ADE-based settings), and comparisons are made against relevant unsupervised and training-free/no-free baselines.
Quantitative results consistently show that TACO significantly outperforms prior unsupervised methods and even approaches the performance of some trained alternatives, which supports the main empirical claims. Qualitative visualizations further illustrate that the learned factors correspond to meaningful semantic regions and align well with the provided audio prompts.

However, there are minor concerns regarding clarity and reporting:
1. Some tables report unusually large standard deviations that seem inconsistent with the deterministic nature of the inference procedure, suggesting adding statistical significance analyses to strengthen the empirical claims.
2. Runtime and computational overhead are discussed but could be quantified more rigorously. Although the method is training-free, Figure 8 indicates that the iterative co-NMF optimization leads to increased inference time. An analysis of how inference cost scales with problem size (e.g., number of image patches, feature dimensionality, or factor rank) would help clarify its practical applicability.
Overall, the evidence is convincing and largely supports the claims.

**Requested Changes:**

1. Clarify and correct numerical reporting in experimental tables: Some reported standard deviations appear unusually large and may result from formatting or alignment issues. The authors should carefully verify all numerical entries, ensure consistent reporting of mean ± standard deviation, and clarify the number of runs used to compute these statistics. Suggesting adding statistical significance analyses to strengthen the empirical claims.
2. Provide a more detailed and broader analysis of inference-time efficiency: Since the method relies on iterative co-NMF optimization at inference time, the paper should report more setup information (eg. memory usage), and iteration counts on representative hardware, along with comparisons to baseline methods.
3. Improve reproducibility scaling details: The authors should clearly specify all hyperparameters (e.g., NMF rank, regularization weights, iteration limits, initialization strategy) and provide a concrete plan to facilitate replication.

---

> ### Author Response · Authors · 2026-01-27
> **Authors answer**
>
> We thank the reviewer for their detailed feedback. We provide here a detailed answer to the raised concerns.
>
> **Clarify and correct numerical reporting in experimental tables: Some reported standard deviations appear unusually large and may result from formatting or alignment issues. The authors should carefully verify all numerical entries, ensure consistent reporting of mean ± standard deviation, and clarify the number of runs used to compute these statistics. Suggesting adding statistical significance analyses to strengthen the empirical claims.**
>
> -> We confirm that the reported values are correct. While some standard deviation are indeed relatively large, this behavior is expected and stems from the random initialization of the matrices $V_A$ and $VI$. To improve clarity, we have revised the manuscript to explicitly state the number of runs used to compute the reported statistics (three runs). This information is now provided at the end of the Implementation Details paragraph in Section 4. In addition, we have incorporated statistical significance tests to assess the superiority of our method over the strongest competing baseline.
>
>
> **Provide a more detailed and broader analysis of inference-time efficiency: Since the method relies on iterative co-NMF optimization at inference time, the paper should report more setup information (eg. memory usage), and iteration counts on representative hardware, along with comparisons to baseline methods.**
>
> -> We have expanded the manuscript to provide a more comprehensive analysis of inference-time efficiency. Specifically, we now report the memory footprint required for the co-NMF decomposition in Section 4.4 ( $\sim$ 425MB for an image of definition 764x764 and a 10 second audio). We also explicitly state the hardware used for all inference-time measurements (NVIDIA A100 GPU). Furthermore, the appendix includes a comparison of encoder computational costs for both our method and the baselines, reported in terms of MACs.
>
> **Improve reproducibility scaling details: The authors should clearly specify all hyperparameters (e.g., NMF rank, regularization weights, iteration limits, initialization strategy) and provide a concrete plan to facilitate replication.**
>
> -> We agree that reproducibility is important, hence all hyperparameters, including NMF rank, regularization coefficients, iteration limits, and initialization strategies, are fully specified in the Implementation Details paragraph of Section 4. To further facilitate reproducibility, we also state that the full codebase will be released upon publication, as noted in the final contribution listed in the Introduction.

---

### Review · Reviewer_UpGK · 2025-12-30

**Summary Of Contributions:**

## Summary
This paper introduces TACO, a training-free framework for sound-prompted segmentation. The method addresses the challenge of identifying image regions corresponding to sounds using only frozen, pre-trained audio (CLAP) and image (CLIP) models. The core technical contribution is a Semantically Constrained Non-negative Matrix Co-Factorization (Sem co-NMF), which identifies shared interpretable concepts across modalities by projecting deep features into a unified semantic space using a word bank as anchors. These identified factors are then used to prompt an open-vocabulary segmenter (FC-CLIP) for high-quality mask generation. Experimental results across five datasets demonstrate that TACO achieves state-of-the-art performance in unsupervised settings, significantly outperforming previous training-based unsupervised methods. Overall, I have a positive view of this paper.


## Strengths
- The paper is well-structured and clearly written, with helpful visualizations and mathematical derivations.
- Most research work on sound-prompted segmentation focus on the supervised setting, which the pixel-level masks are required. This paper explores a training-free approach, which is valuable and interesting.

## Weaknesses
-  The method relies heavily on a "word bank" to act as semantic anchors. While the authors ablate this using AudioSet tags, there is a risk that the performance is sensitive to the vocabulary size and relevance. If a sound occurs that is not well-represented in the word bank, the "semantic descriptors" might become noisy, leading to a breakdown in the co-factorization.
- class inference in Semantic Segmentation. As admitted by the authors, the semantic segmentation performance on AVSS is limited because the model collapses multiple objects into a single class label during decoding. If an image has two distinct sounding objects (e.g., a dog and a person), TACO segments both but likely assigns them both the same label, which is a significant architectural limitation for true semantic segmentation.
- While the authors state the optimization overhead is minimal, requiring 1000-1800 iterations of gradient descent per sample at inference time is a disadvantage compared to a single-pass feed-forward network.
- Failure in multi-source scenarios: qualitative analysis (Figure 13) shows that the model still struggles to encode multiple distinct audio concepts effectively, often neglecting one or providing partial results. This suggests the "min" operator in the penalty function might be too aggressive, focusing only on the single strongest shared concept.
- The related work in current manuscript primarily focuses on the sound-prompted segmentation, which focuses on spatial-level audiovisual understanding. Introducing more relevant contexts on a broader audio-visual learning, such as temporal learning, would make the manuscript more comprehensive.

**Audience:**

Yes

**Audience Explanation:**

This paper focuses on the sound-prompted segmentation, which may be attracted to audiovisual learning and segmentation community.

**Broader Impact Concerns:**

Not required. But it would be better to add a paragraph to claim the Broader Impact Statement.

**Claims And Evidence:**

Yes

**Claims Explanation:**

The proposed method is clearly elaborated and it makes sound. The authors verify the method on standard datasets, which demonstrate the its effectiveness and superiority.

**Requested Changes:**

Please see the weakness part.

---

> ### Author Response · Authors · 2026-01-27
> **Authors answer**
>
> We thank the reviewer for their detailed review. We provide here a detailed answer to the raised concerns.
>
>
> **[...] there is a risk that the performance is sensitive to the vocabulary size and relevance.**
>
> -> We agree that the performance of the method is conditioned on the choice of the word bank. However, we argue that AudioSet tags constitute a sufficiently general and expressive vocabulary, as they cover a wide range of broad and diverse audio concepts.
> We edited the manuscript to add Figure 8 that shows the performance of TACO depending on the number of word in the bank. It shows that half of the word bank can be remove while the performance remains stable. This shows the resilience of the method to the size of the word bank. We hypothesize that this emerges from compositionnality capabilities of the text encoders: concept matched can be interpolation of multiple words in the word bank.
> Finally, we ran an experiment where we replaced the word bank by a set of sounds obtaind by asking GTP5.2 to generates sounds sources. The results show (in Table 4) that performance drops only marginally, highlighting the robustness of the method to the content of the word bank, and not only its size.
>
>
> **[...]If an image has two distinct sounding objects (e.g., a dog and a person), TACO segments both but likely assigns them both the same label, which is a significant architectural limitation for true semantic segmentation.**
>
> -> We agree with the reviewer that the TACO formulation does not fully resolve multi-label semantic segmentation scenarios. We emphasize, however, that this work represents the first attempt to perform semantic segmentation via audio-visual decomposition. As discussed in Section 4.2, addressing multi-label decoding and more fine-grained semantic assignments is an important direction for future work and we argue that this is beyond the scope of the present study. We edited the 'Conclusion and limitations' section in the manuscript to make this limitation clearer.
>
> **[...]requiring 1000-1800 iterations of gradient descent per sample at inference time is a disadvantage compared to a single-pass feed-forward network.**
>
> -> While the iterative optimization does introduce additional computational overhead, our empirical analysis shows that this cost remains moderate relative to the overall inference pipeline. As reported in Figure 9, the co-NMF optimization accounts for only an additional 20–40% computation compared to the cost of the large backbone encoders, depending on the number of optimization steps. Thus, the dominant computational cost remains the encoder forward passes, and concurrent works that rely on more expensive backbones sometimes have overall higher costs (as shown in Table 11).
>
> **Failure in multi-source scenarios [...] suggests the "min" operator in the penalty function might be too aggressive, focusing only on the single strongest shared concept.**
>
> -> We acknowledge that there exist challenging cases in which the model fails to fully capture all sources, as illustrated in the Appendix. However, such cases do not represent the majority of scenarios. As shown qualitatively in Figure 4 (last row) and quantitatively in Table 1, TACO performs robustly in multi-source settings and achieves state-of-the-art results overall.
> To make it clearer, we added to the manuscript an additionnal experiment where we use apply the penalty on the bottom-2 factors instead of only using the min operator (e.g we force two common factors) and present the results in Appendix (Table 9). Enforcing two common factors result in a slight drop of performance, confirming the hypothesis that using the min operator allow to put all the common audio-visual information in the common factor.
>
>
> **[...]Introducing more relevant contexts on a broader audio-visual learning, such as temporal learning, would make the manuscript more comprehensive.**
>
> -> We have revised the related work section to include a broader discussion of audio-visual learning methods beyond sound-prompted segmentation, covering more general paradigms in audiovisual representation learning.

---

### Review · Reviewer_wRnq · 2026-01-09

**Summary Of Contributions:**

This paper proposes TACO, a training-free approach to sound-prompted image segmentation that operates entirely on frozen pretrained audio and visual encoders. The core idea is to jointly decompose audio and visual token representations using a semantically constrained audio–visual co-NMF objective, which reveals shared and interpretable latent factors across modalities. A semantic penalty based on text-anchored descriptors aligns the decompositions, enabling the method to identify the dominant “sounding” factor. This factor is then used both as a coarse segmentation mask and as a prompt to an open-vocabulary segmentation model for refined and semantic segmentation.

Key strengths include the genuinely training-free formulation, the interpretability of the factorization-based approach, and strong empirical performance across several sound-prompted segmentation benchmarks. Notable weaknesses include inference-time optimization cost and limitations in multi-class semantic decoding, which are acknowledged but not fully resolved.

**Additional Comments:**

Overall, this is a solid and well-executed paper with a clear conceptual contribution and strong empirical support. While there are practical limitations related to inference-time optimization and semantic decoding, these do not undermine the core idea. With minor clarification and framing improvements, the paper would make a valuable contribution to the literature on training-free and unsupervised multimodal learning.

**Audience:**

Yes

**Audience Explanation:**

The paper will likely be of interest to researchers working on multimodal learning, audio–visual understanding, unsupervised representation learning, and methods that leverage frozen foundation models. The training-free perspective and factorization-based formulation provide a useful alternative to data-hungry training pipelines, and the approach may inspire related work in other cross-modal or weakly supervised settings. As such, the findings are relevant to a meaningful subset of TMLR’s readership.

**Broader Impact Concerns:**

I do not see significant broader impact or ethical concerns specific to this work. The method relies on standard pretrained models and is intended for general computer vision and audio–visual understanding tasks. As with related segmentation techniques, potential misuse would fall under broader concerns already associated with visual recognition systems, rather than arising uniquely from this contribution.

**Claims And Evidence:**

Yes

**Claims Explanation:**

The paper provides clear and generally convincing evidence in support of its claims. The method is evaluated across multiple benchmarks covering both binary and semantic sound-prompted segmentation, and results are consistently competitive or superior to prior unsupervised or training-based baselines. The experimental section includes meaningful ablations on semantic constraints, number of factors, word-bank size, and descriptor design, which help isolate the contributions of different components. Qualitative visualizations further support the interpretability claims by showing that the selected factors correspond well to sounding objects.

While inference-time optimization introduces additional complexity, runtime behavior is analyzed and performance saturation is empirically demonstrated. Overall, the empirical evidence is sufficient to support the central claims of the paper.

**Requested Changes:**

1. **Clarify inference-time optimization behavior (non-critical).**

   Provide additional discussion or statistics on convergence stability, sensitivity to initialization, or adaptive stopping criteria, given the reliance on per-sample optimization.

2. **Expose semantic-anchor sensitivity more clearly (non-critical).**

   The appendix shows performance degradation when the word bank is aggressively reduced. Briefly summarizing this sensitivity in the main paper would clarify underlying assumptions.

3. **Address multi-class semantic decoding limitations (non-critical).**

   Consider a simple extension or baseline that decodes multiple factors in multi-source scenes to assess whether the current limitation is fundamental or largely a design choice.

4. **Sharpen the “training-free” framing (editorial).**

   Explicitly distinguish between the absence of dataset training and the use of inference-time optimization to avoid potential misinterpretation.

---

> ### Author Response · Authors · 2026-01-27
> **Authors answer**
>
> We thank the reviewer for their remarks, which allow to improve the quality of the paper.
>
> **Clarify inference-time optimization behavior (non-critical):
> Provide additional discussion or statistics on convergence stability, sensitivity to initialization, or adaptive stopping criteria, given the reliance on per-sample optimization.**
>
> -> We address sensitivity to initialization by reporting results over multiple independent runs (three) and by providing the standard deviation across all experiments, in Section 4. In addition, we analyze performance as a function of the number of optimization steps, which is reported in Figure 9. Together, these results illustrate the stability of the optimization process and its (early) convergence behavior.
>
> **Expose semantic-anchor sensitivity more clearly (non-critical).
> The appendix shows performance degradation when the word bank is aggressively reduced. Briefly summarizing this sensitivity in the main paper would clarify underlying assumptions.**
>
> -> We have revised the manuscript to include a figure in Section 4 (Figure 8) that explicitly shows performance degradation as a function of the word bank size: its size can be divided by two without significant degradation of the performances. Additionnaly, we generated another work bank using GTP-5.2 and measured TACO's performance using it, in Table 4. Performance drops only marginally, highlighting the robustness of the method to the content of the word bank.
>
> **Address multi-class semantic decoding limitations (non-critical).
> Consider a simple extension or baseline that decodes multiple factors in multi-source scenes to assess whether the current limitation is fundamental or largely a design choice.**
>
> -> As discussed in the manuscript, semantic segmentation is not the primary objective of this work but rather an emergent property of the proposed audio-visual decomposition framework. We therefore did not introduce additional architectural extensions in this work. We do acknoweledge that investigating explicit multi-class decoding strategies to better handle multi-source scenes is an interesting direction for future research. We edited the conclusion of the manuscript to make this limitation clearer.
>
>
> **Sharpen the “training-free” framing (editorial).
> Explicitly distinguish between the absence of dataset training and the use of inference-time optimization to avoid potential misinterpretation.**
>
> -> We agree that it is important to clarify this point as it is a key feature of TACO, and thanks the reviewer for highlighting this point. We have revised the manuscript to explicitly clarify this distinction at the end of the Introduction, emphasizing that the method does not require dataset-level training but does rely on per-sample inference-time optimization.

---

### Review · Reviewer_MjHF · 2026-01-15

**Summary Of Contributions:**

This paper introduces TACO, a training-free approach for audio-visual segmentation that operates entirely on frozen representations from pretrained models (CLIP for images and CLAP for audio). The method performs a semantically constrained audio–visual co-factorization at inference time to discover latent factors that are jointly activated by both modalities. To bridge the mismatch between CLIP and CLAP embedding spaces, TACO uses a semantic anchor word bank to align audio and visual factors via a dedicated penalty. The most strongly aligned “sounding factor” is then used to produce a segmentation, either directly from the factor activations or by prompting an open-vocabulary segmenter (FC-CLIP) with the discovered image factor. Experiments across multiple benchmarks show that TACO achieves promising performance.

The underlying problem formulation is not new, and there exists prior zero-shot audio-visual segmentation approach that leverages CLAP together with pretrained grounding and segmentation models (e.g., Grounding-DINO and SAM). In this context, the primary contribution of this work lies in its use of a soft co-NMF framework, with textual semantic anchors serving as a bridge, to associate audio and visual representations in a training-free manner for audio-visual segmentation.

**Audience:**

Yes

**Audience Explanation:**

Yes. The paper would interest part of the TMLR audience, particularly researchers in multimodal learning,  training-free methods, and foundation model reuse. It demonstrates that frozen pretrained audio–visual representations can be aligned via semantic factorization to achieve strong segmentation performance offers a novel and relevant perspective for the community.

**Claims And Evidence:**

No

**Claims Explanation:**

- The central motivation of this paper is to answer the question: Can sound-prompted visual segmentation be achieved using only pretrained audio and image models, in an unsupervised manner? The authors argue that prior work largely relies on carefully designed architectures and task-specific training procedures. However, to the best of my knowledge, there already exist works that address this problem without additional training. For example, Yu et al. [1] explored a zero-shot formulation of audio-visual segmentation as early as 2023, using CLAP to capture semantic concepts from audio and then prompting a segmentation model. The insight is conceptually close to the motivation of this paper. More recently, Chen et al. [2] proposed a fully training-free, open-vocabulary audio-visual segmentation framework using foundation models. These prior works are not discussed in the paper, making the claim of being “the first training-free approach for sound-prompted segmentation” inaccurate.

- Regarding the proposed approach, it is unclear how the method handles multiple instances of the same object category. For example, if multiple guitars appear in the video but only one is producing sound, the discovered image factor—used as a global prompt—may lead the model to segment all guitars, including non-sounding instances. This raises concerns about correct instance-level localization.

- The method’s performance and efficiency appear to be sensitive to the choice and size of the vocabulary used as semantic anchors. A large vocabulary may significantly increase optimization time during inference, potentially limiting scalability. A small one will definitely drop the model performance.

- The approach primarily selects a single best-matching factor to represent the sounding concept.  The authors provided results on the MS3 set, which contains multi-source videos.  However, it remains unclear how the method explicitly handles multiple simultaneous sound sources, beyond implicitly encoding them within a single factor, which may limit its applicability in more complex auditory scenes.

[1] Yu, Jiarui, Haoran Li, Yanbin Hao, Jinmeng Wu, Tong Xu, Shuo Wang, and Xiangnan He. "How Can Contrastive Pre-training Benefit Audio-Visual Segmentation? A Study from Supervised and Zero-shot Perspectives." In BMVC, pp. 367-374. 2023.

[2]Chen, Shengkai, Yifang Yin, Jinming Cao, Shili Xiang, Zhenguang Liu, and Roger Zimmermann. "OpenAVS: Training-Free Open-Vocabulary Audio Visual Segmentation with Foundational Models." arXiv preprint arXiv:2505.01448 (2025).

**Requested Changes:**

I hope the authors can adequately address the concerns raised above and make the necessary revisions to strengthen the paper.

---

> ### Author Response · Authors · 2026-01-27
> **Authors answer**
>
> We thank the reviewer for their feedback. We provide here a detailed answer to the raised concerns.
>
> **[...] Prior works are not discussed in the paper, making the claim of being “the first training-free approach for sound-prompted segmentation” inaccurate.**
>
>
> -> We thank the reviewer for pointing out these relevant works and agree that they should be discussed for proper contextualization. We emphasize, however, that these approaches differ fundamentally from ours in both formulation and technical contribution.
>
> We wish to remind that our method aims at explicitly discovering shared audio-visual factors through co-factorization. In fact, as stated in the introduction our goal is to 'enables finding audio-visual correspondences across unaligned sound and image representation backbones'.
>
> Regarding Chen et al. [2], their framework relies on a cascade of large and heterogeneous models, including LLMs (e.g., GPT-4o-mini, SAM2, GroundingDINO, Qwen-omni, and Pengi), which makes it computationally heavy and difficult to deploy in practice. In contrast, our approach relies solely on pretrained audio–visual encoders and a lightweight optimization procedure, resulting in a substantially simpler and more efficient pipeline.
> We want to highlight that this paper, to our knowledge, is currently a preprint and has not yet been published in a conference or journal. Nevertheless, we decided to include it in the related work section as we consider it a valuable reference for interested readers. However, we did not include it in our experiments because, in addition to its preprint status, we believe it would mislead the interpretation of the results for several reasons:
>
> * The reliance on external APIs makes a direct comparison of computation and latency challenging.
> * The ablation studies included in [2] (see Tab. 3 in the ICLR submission version, which is more recent than the arXiv version) show that the high performance is mostly due to the use of GPT-4o-mini and SAM2. On the S4 dataset, using publicly available models (GPT2-XL and SAM) leads to an mIoU of 0.431 vs. 0.604 for TACO. Therefore, reporting this open-source baseline would be irrelevant as it is far from the state of the art. Conversely, reporting their best-performing model (OpenAVS-Large) would require substantial explanations that could obscure the results and are tangential to the audio-visual alignment problem addressed in this work.
> * There is a significant risk of data leakage when using models trained on undisclosed data. While we use OpenCLIP and CLAP with open training data, the method in [2] relies on GPT-4o-mini and Qwen-omni, which may have seen the S4 or MS3 datasets during their training. This is particularly likely given that both S4 and MS3 are composed of YouTube videos.
>
>
> Yu et al. [1] propose a training-free AVS method that similarly does not explicitly model audio–visual correspondences. Their approach is closely related to our A-FCCLIP baseline, as it essentially prompts SAM using audio-derived class labels from CLAP, which are first localized in the image using GroundingDINO. Another key difference lies in their multi-source setting, where the number of sources is explicitly fixed (e.g., enforcing two sources), providing strong prior information that partially explains the reported performance gains in that specific setup. We agree that this comparison is important and have included it in Table 1. Additionally, we revised the manuscript to clarify the claims regarding training-free methods in the Introduction.
>
>
> **Regarding the proposed approach, it is unclear how the method handles multiple instances of the same object category.**
>
> -> We acknowledge that the proposed method does not explicitly address instance-level disambiguation when multiple visually similar objects are present but only one is sounding. Importantly, this limitation is shared with most of other sound-prompted approaches, including the baselines we are comparing to. These approaches primarily rely on global semantic alignment or class-level correspondence and do not incorporate explicit audio-to-instance binding mechanisms. As a result, instance-level localization in such scenarios remains an open challenge in audio-visual segmentation. Addressing this limitation would likely require dedicated instance-aware modeling or temporal reasoning, which we consider an important direction for future work.

---

> > ### Author Response · Authors · 2026-01-27
> >
> > **The method’s performance and efficiency appear to be sensitive to the choice and size of the vocabulary used as semantic anchors.**
> >
> > -> We agree that the choice of vocabulary size affects both performance and computational cost.
> > However, we first argue that the  amount of additionnal computation induced by the distances to the word bank is negligible (~13% of the MACs of the decomposition when using the full general word bank), as the decomposed matrices shape do not depend on it.
> > Moreover, as shown in Section 4.4, even when using the full word bank, the total inference-time optimization remains efficient, accounting for approximately 30% additional computation relative to the encoder forward passes.
> >
> > Furthermore, our ablation study in the appendix shows that reducing the word bank size by half results in only a modest performance drop (−1.1 Mask-IoU on S4). To further clarify this trade-off, we have added a new figure in the Section 4 (Figure 8) that explicitly reports performance as a function of the word bank size. These results demonstrate that the method degrades gracefully and remains robust across a wide range of vocabulary sizes.
> > Finally, added an experiment where we replaced the word bank by a set of sounds obtaind by asking GTP-5.2 to generates sounds sources. The results show (in Table 4) that performance almost do not drop, highlighting the robustness of the method to the content of the word bank, and not only its size.
> >
> > **It remains unclear how the method explicitly handles multiple simultaneous sound sources**
> >
> > -> As illustrated in Figure 5 and explained in the accompanying discussion, the proposed co-factorization does not encode a single dominant concept but instead captures all sounding objects within the shared audio-visual factor.
> > We argue that this capability is enabled because the decomposition matches factors and not concepts from the word bank, hence they can be more abstracts and contain more than single concepts.
> > As a result, the learned factor can represent multiple concurrent sources rather than collapsing to a single dominant one, which is corroborated by both qualitative examples and quantitative results on multi-source benchmarks.
> > Finally, we edited the manuscript to add an additionnal experiment where we use apply the penalty on the bottom-2 factors instead of only using the min operator (e.g we force two common factors) and present the results in Appendix (Table 9). Enforcing two common factors result in a slight drop of performance, confirming the hypothesis that using the min operator allow to put all the common audio-visual information in the common factor.

---

### Decision · Action_Editor_N2gD · 2026-02-16

**Recommendation:** Accept as is

**Audience:**

Yes

**Audience Explanation:**

All reviewers agreed that the paper is of interest to a portion of the TMLR audience, even prior to the revision and discussion.

**Claims And Evidence:**

Yes

**Claims Explanation:**

All four reviewers agree that the claims made in the revised manuscript are supported by accurate, convincing, and clear evidence.

The revision process led to substantial improvements over the original submission in the following areas.

**Positioning with respect to prior work.** The original manuscript incorrectly claimed to be the first paper to investigate training-free audio-prompted segmentation. The revision is correctly positioned as studying a new approach based on semantic co-factorization of frozen audio and image representations while also citing the prior work pointed out by Reviewer MjHF.

**Sensitivity to choice and size of vocabulary of semantic anchors.** (Raised by reviewers MjHF, wRnq, UpGK) The revision adds a number of results illustrating the influence of the size of the semantic anchor set and its content on the performance of the method.

**Cost of inference.** (Raised by reviewers wRnq, UpGK, MPy6) The revision adds an analysis of the inference-time optimization compute and memory requirements and shows the relationship between the performance of the method and the number of iterations of optimization.

**Multiple sound sources.** (Raised by reviewers MjHF, wRnq, UpGK) The revision emphasizes that TACO encodes all sounding factors into a single concept, and thus does not separate out multiple simultaneous sound sources. This issue is identified as a limitation of the current approach. Similarly, the authors state that the current approach does not handle multiple instances of the same object category.

**Large standard deviations in some tables are worrying; should add statistical significance analysis.** (Raised by reviewers MPy6, wRnq) In the discussion the authors clarified that the variability is due to the random initialization of the $U_{I}$ and $U_{A}$ matrices, and the revision states this explicitly in the Results and Discussion.

**Improve reproducibility.** (Reviewer MPy6) All hyperparameters are specified in the implementation details and the authors promised to release code.